# An image-computable model of speeded decision-making

**Paul I Jaffe[1]\*, Gustavo X Santiago-Reyes[2], Robert J Schafer[3], Patrick G Bissett[1], Russell A Poldrack[1]**

[1]Department of Psychology, Stanford University, Stanford, United States; [2]Department of Bioengineering, Stanford University, Stanford, United States; [3]Lumos Labs, San Francisco, United States

## eLife Assessment

This **important** study presents an original and promising approach to combine convolutional neural networks of visual processing with evidence accumulation models of decision-making. While the methodological approach is technically sophisticated and the evidence is **solid**, there is still a gap between the model and the behavioral data. The study will be of interest to researchers working in the fields of machine learning and cognitive modeling.

**\*For correspondence:**
pijaffe@stanford.edu

**Abstract** Evidence accumulation models (EAMs) are the dominant framework for modeling response time (RT) data from speeded decision-making tasks. While providing a good quantitative description of RT data in terms of abstract perceptual representations, EAMs do not explain how the visual system extracts these representations in the first place. To address this limitation, we introduce the visual accumulator model (VAM), in which convolutional neural network models of visual processing and traditional EAMs are jointly fitted to trial-level RTs and raw (pixel-space) visual stimuli from individual subjects in a unified Bayesian framework. Models fitted to large-scale cognitive training data from a stylized flanker task captured individual differences in congruency effects, RTs, and accuracy. We find evidence that the selection of task-relevant information occurs through the orthogonalization of relevant and irrelevant representations, demonstrating how our framework can be used to relate visual representations to behavioral outputs. Together, our work provides a probabilistic framework for both constraining neural network models of vision with behavioral data and studying how the visual system extracts representations that guide decisions.

## Introduction

Decision-making under time pressure is deeply embedded within the activities of daily life. To study the cognitive and neural processes underlying decision-making, psychologists fit computational models to response time (RT) data gathered from relatively simple cognitive tasks. Evidence accumulation models (EAMs), such as the diffusion-decision model (*Ratcliff and McKoon, 2008*; *Ratcliff, 1978*) and linear ballistic accumulator model (LBA) (*Brown and Heathcote, 2008*), are the most successful and widely-used computational models of joint RT and choice data from decision-making tasks. In the EAM framework, sensory evidence is accumulated (possibly with noise) until reaching a threshold, at which point an overt response is generated. As such, decision-making is described in terms of a small number of interpretable parameters that capture basic cognitive processes. Empirically, EAMs have been shown to capture RT distributions from a variety of cognitive tasks at the individual subject level (*Brown and Heathcote, 2008*; *Ratcliff, 1978*; *Usher and McClelland, 2001*; *Ratcliff and Rouder, 1998*).

Despite this empirical success and theoretical merit, the simple characterization of decision-making encapsulated by EAMs results in somewhat restrictive applications (*Evans and Wagenmakers, 2020*). In the context of visual tasks, EAMs do not typically provide a detailed specification of how raw visual stimuli are transformed into evidence and consequently do not explain the visual processing steps underlying decision-making. A number of recent modeling efforts have addressed this limitation by adapting convolutional neural network (CNN) models used in image classification tasks to the novel purpose of generating RTs or RT proxies (*Spoerer et al., 2020*; *Taylor et al., 2021*; *Kumbhar et al., 2020*; *Rafiei et al., 2024*; *Goetschalckx et al., 2023*; *Annis et al., 2021*; *Holmes et al., 2020*; *Trueblood et al., 2021*), a strategy we also pursue here. CNNs are useful in this regard since they are image-computable—they accept arbitrary images as input—and capture important characteristics of biological vision (*Lindsay, 2021*; *Yamins and DiCarlo, 2016*; *Kriegeskorte, 2015*). Early CNN layers exhibit spatially-organized units with local receptive fields, analogous to the retinotopic organization of the early mammalian visual pathway. The concatenation of multiple layers forms a hierarchy in which progressively more abstract features are extracted in deeper layers, in a way that is globally similar to the primate visual cortical hierarchy.

Here, we integrate CNN models for visual feature extraction with traditional EAMs of speeded decision-making tasks in a framework we call the visual accumulator model (VAM). As in the prior models referenced above, the VAM accepts raw (pixel-space) visual stimuli as inputs and generates RTs and choices as outputs. The key feature of the VAM that distinguishes it from prior models is that the CNN and EAM parameters are *jointly fitted* to the RT, choice, and visual stimulus data from individual participants in a unified Bayesian framework. Thus, both the visual representations learned by the CNN and the EAM parameters are directly constrained by behavioral data. In contrast, prior models first optimize the CNN to perform the behavioral task, then separately fit a minimal set of high-level CNN parameters (*Rafiei et al., 2024*) and/or the EAM parameters to behavioral data (*Annis et al., 2021*; *Holmes et al., 2020*; *Trueblood et al., 2021*). As we will show, fitting the CNN with human data—rather than optimizing the model to perform a task—has significant consequences for the representations learned by the model.

We leverage the VAM to explore how abstract, task-relevant information is extracted from raw sensory inputs, and we investigate how the behavioral phenomenon of congruency effects arises as a consequence of the representation geometry learned by the CNN. In doing so, our framework also addresses one of the main criticisms of deep neural network models of vision that are optimized to perform particular tasks (e.g. object identification): these models do not account for many results from psychology (*Baker et al., 2018*; *Bowers et al., 2022*; *Fel et al., 2022*; *Lindsay, 2021*; *Malhotra et al., 2022*).

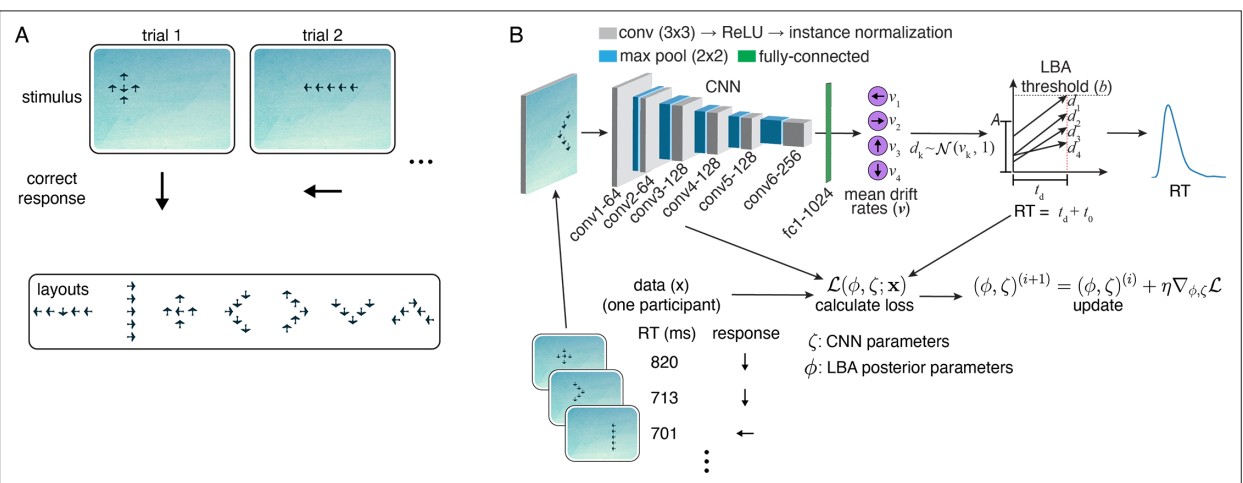

**Figure 1.** Task and model. (**A**) Top, Lost in Migration task. Bottom, the seven stimulus layouts (random target/flanker directions). (**B**) Visual accumulator model (VAM) schematic. The numbers after the convolutional neural network (CNN) layer names correspond to the number of channels used in that layer. See Methods for additional details.

## Modeling framework

We first describe the task—Lost in Migration (LIM), available as part of the Lumosity cognitive training platform—which will serve to ground the discussion of the model. LIM is a stylized version of the well-known arrows flanker task used in the study of cognitive control and visual attention (*Figure 1A*; *Stoffels and van der Molen, 1988*). In LIM, participants are shown stimuli composed of several arrow-shaped birds. The task is to indicate the direction of the central bird (the target) using the arrow keys on the keyboard, ignoring the direction of the surrounding birds (the flankers). Participants engage with the task at home and are rewarded via a composite score that takes into account both speed and accuracy.

The stimuli used in LIM vary along several dimensions that are not present in the standard flanker task, implying potentially complicated stimulus-behavior dependencies that are well-suited to the VAM. First, both targets and flankers can be independently oriented left, right, up, or down, such that there are four possible response directions (the standard arrow flanker task allows for only left and right responses). Second, the layout of the targets and the flankers can appear in one of seven configurations: left/right/up/down 'V,' horizontal/vertical line, and cross (*Figure 1A*). Last, the target can be centered anywhere within the 640×480 pixel game window, subject to edge constraints.

As with other flanker tasks, a given trial is said to be congruent if targets and flankers are oriented in the same direction, and incongruent otherwise. A consistent observation from studies employing the flanker task and related conflict tasks are *congruency effects*: participants are slower and less accurate on incongruent trials (*Eriksen and Eriksen, 1974*; *Simon, 1982*; *Stroop, 1935*). Congruency effects are considered to index a specific aspect of cognitive control: they indicate the extent to which an individual can selectively attend to task-relevant information and ignore irrelevant information.

The visual accumulator is an image-computable EAM, composed of a CNN and EAM chained together (*Figure 1B*). Minimally processed (pixel-space) visual stimuli are provided as inputs to the CNN. The outputs of the CNN correspond to the mean rates at which evidence is accumulated (the drift rates), one for each possible response (four in the case of LIM). The EAM then generates choices and RTs through a noisy evidence accumulation process. For each participant in the dataset, we fit one such model (CNN + EAM) jointly using that participant's visual stimuli, RTs, and choices. Building on prior work (*Dao et al., 2024*; *Kucukelbir et al., 2017*), we developed an automatic differentiation variational inference (ADVI) algorithm that simultaneously optimizes the parameters of the CNN and learns the posterior distribution over the LBA parameters (Methods).

The particular EAM we adopt is the linear ballistic accumulator model (LBA *Brown and Heathcote, 2008*; *Figure 1B*) since it can be applied to tasks with more than two possible responses, though the general VAM framework is compatible with other EAMs that have a closed-form or easily approximated likelihood. The LBA parameters fitted in the VAM are the decision threshold $b$, the non-decision time $t_0$, and a parameter $A$ that controls the dispersion of the initial accumulator values (the drift rate means are fitted implicitly via the CNN). We used a seven layer CNN architecture (six convolutional layers, one fully-connected layer) in the VAM (*Figure 1B*). To speed up the training process, the first two convolutional layers were initialized with the parameters from a 16-layer VGG CNN trained to classify the ImageNet dataset (*Deng et al., 2009*; *Simonyan and Zisserman, 2015*).

All of the code (https://github.com/pauljaffe/vam, copy archived at *Jaffe, 2025*) and data (https://doi.org/10.5281/zenodo.10775513) used to train the VAMs and reproduce our results are publicly available.

## Results

### Models capture human behavioral data

We fitted a separate VAM to LIM data (visual stimuli, RTs, choices) from each of 75 Lumosity users (participants). We selected participants who had practiced Lost in Migration extensively (≥25,000 trials) and used data from a later stage of practice to minimize learning effects (≥ each participant's 50th gameplay). These participants varied in age (23–87 y) and in their behavior (mean RT, accuracy, congruency effects), allowing us to examine how well our framework captured individual differences.

To assess the model fits, we compared the mean RT, accuracy, RT congruency effect (incongruent trial mean RT minus congruent trial mean RT), and accuracy congruency effect (congruent trial accuracy minus incongruent trial accuracy) of each model/participant pair on a set of holdout stimuli, separate

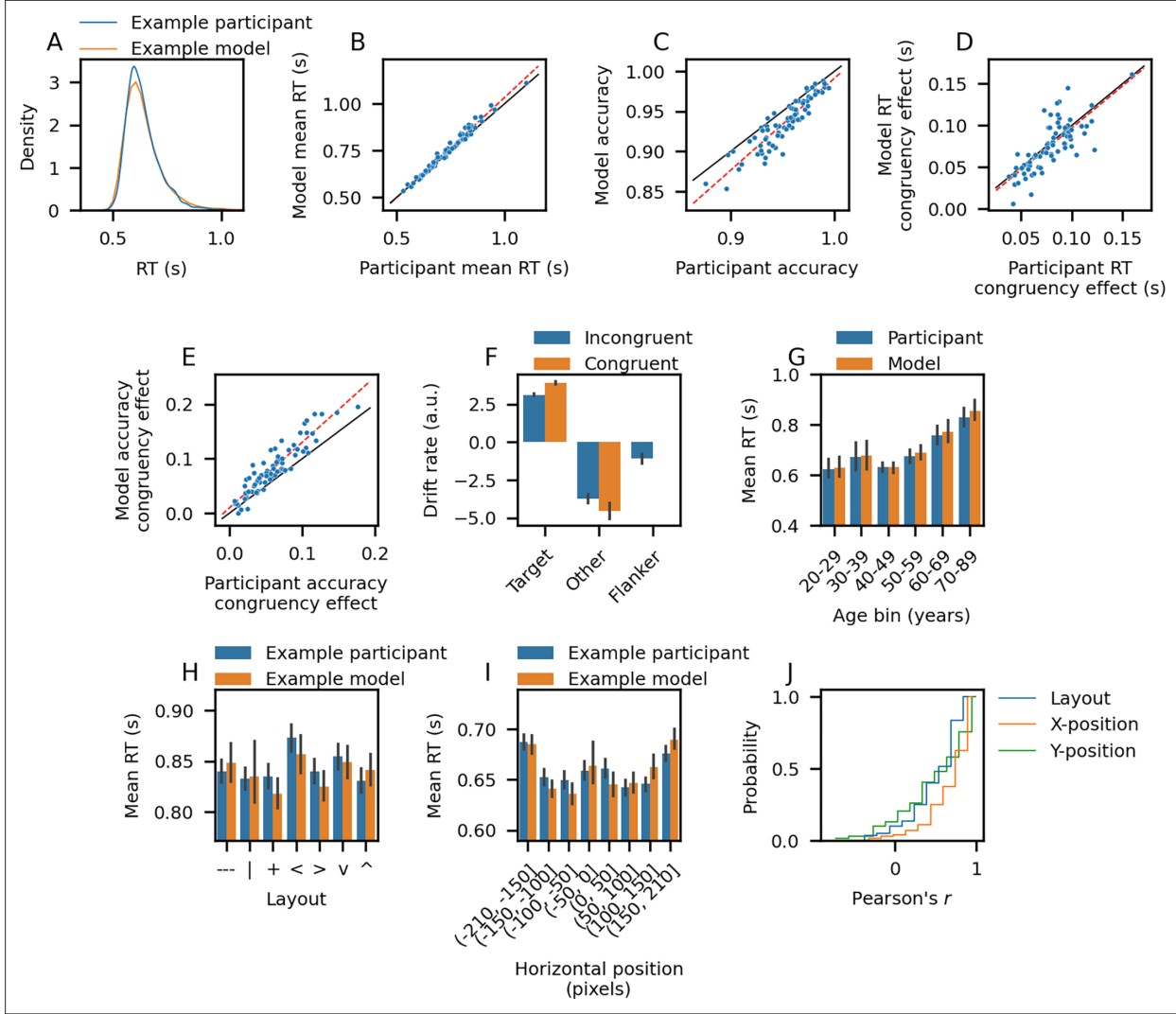

**Figure 2.** Comparison of model/participant behavior. For panels **B–E**, each point is one model/participant (n=75), black line: unity, red line: linear best fit. (**A**) Example model/participant response time (RT) distributions. (**B**) Mean RT (Pearson's $r$ = 0.99, bootstrap 95% CI = (0.99, 0.99), best fit slope = 1.07). (**C**) Accuracy ($r$=0.91, 95% CI = (0.87, 0.94), slope = 1.15). (**D**) RT congruency effect ($r$=0.77, 95% CI = (0.67, 0.86), slope = 1.01). (**E**) Accuracy congruency effect ($r$=0.92, 95% CI = (0.88, 0.94), slope = 1.20). (**F**) Drift rates averaged across all trials and models. (**G**) Mean RT vs. age averaged across models. (**H**) Example model/participant mean RT vs. stimulus layout (Pearson's $r$ = 0.67). (**I**) Example model/participant mean RT vs. horizontal stimulus position (negative values: left of center; Pearson's $r$ = 0.79). (**J**) Empirical CDF of Pearson's $r$ between model/participant mean RTs across stimulus feature bins (only participants with significant RT modulation are shown; layout: $n$ = 60 models/participants, x-position: $n$ = 72, y-position: $n$ = 69). Error bars in panels (**F-I**) are bootstrap 95% confidence intervals.

The online version of this article includes the following figure supplement(s) for figure 2:

**Figure supplement 1.** Example model/participant response time (RT) distributions and dependence of RTs on stimulus features.

**Figure supplement 2.** Age dependence of linear ballistic accumulator (LBA) parameters.

**Figure supplement 3.** Dependence of response times (RTs) on stimulus layout and position.

**Figure supplement 4.** Response time (RT) delta plots and conditional accuracy functions.

from those used to train the models (*Figure 2* and *Figure 2—figure supplement 1*). For each behavioral summary statistic, the responses of the fitted models were highly correlated with those of the participants (Pearson's $r > 0.75$), with slopes close to unity (*Figure 2B–E*, statistics in figure legend). We found that the RT congruency effect could be attributed to a reduction in the mean target drift rate parameter on incongruent vs. congruent trials, while the accuracy congruency effect could be attributed to a higher mean flanker drift rate on incongruent trials relative to the non-target (other) mean drift rates on congruent trials (*Figure 2F*). The latter observation follows from the fact that the drift rates

on trial $i$ are sampled from $\mathcal{N}(v^{(i)}, 1)$, where the $v^{(i)}$ are the mean drift rates shown in **Figure 2F**. Since the flanker drift rates on incongruent trials are higher (less negative) than the non-target drift rates on congruent trials, more errors result on incongruent trials, giving rise to the accuracy congruency effect.

We also examined whether the fitted models captured demographic effects, focusing on the well-established slowing of RTs that occurs with age (**Gottsdanker, 1982**; **Nettelbeck and Rabbitt, 1992**; **Ratcliff, 1978**). Mean RTs began increasing around age 50, effects captured by the fitted models (**Figure 2G**). Consistent with prior work in a variety of decision-making tasks (**Forstmann et al., 2011**; **Ratcliff et al., 2001**; **Servant and Evans, 2020**; **Steyvers et al., 2019**), we found that an age-dependent increase in the non-decision time ($t_0$) component of the response contributed to longer RTs in the models from older adults (**Figure 2—figure supplement 2**). In contrast to these prior studies, we did not observe increased response caution (measured as $b - A$) in the models from older adults (**Figure 2—figure supplement 2**). This discrepancy could be explained by differences in the task design or the fact that our participants had practiced the task substantially more than in typical studies (**Steyvers et al., 2019**). We also observed an age-dependent reduction in the target drift rates and no age-dependence of the flanker drift rates (**Figure 2—figure supplement 2**), findings that have received mixed support in the literature (**Ben-David et al., 2014**; **Forstmann et al., 2011**; **Ratcliff, 1978**; **Servant and Evans, 2020**; **Steyvers et al., 2019**).

One virtue of the VAM is that the model implicitly learns which stimulus properties influence behavior. As a simple demonstration of this, we investigated how two high-level visual features influenced participant behavior—stimulus layout and both horizontal/vertical stimulus position—and whether the fitted models captured these effects. We focused on RT effects, since no participants exhibited a significant effect of layout or horizontal/vertical position on accuracy ($p>0.05$ for all stimulus features, chi-squared test), though we note that ceiling effects resulting from the high accuracy of the participants may have hindered our ability to detect these accuracy effects.

The majority of participants exhibited layout-dependent RT biases ($p<0.05$ for 60/75 participants, one-way ANOVA; see examples in **Figure 2H** and **Figure 2—figure supplement 1**). We quantified how well the models captured these effects by calculating the Pearson's $r$ between model/participant mean RTs across each layout for the participants that exhibited significant layout-dependent RT modulation (**Figure 2J**). The median Pearson's $r$ was 0.67, demonstrating good correspondence between model/participant behavior. There was considerable heterogeneity in the particular layout RT biases exhibited by both the participants and models, though responses to trials with the vertical line layout were on average somewhat faster than the other layouts for both participants and models (**Figure 2—figure supplements 1 and 3**).

The majority of participants also exhibited both horizontal and vertical position-dependent RT biases (horizontal position: $p<0.05$ for 72/75 participants, vertical position: $p<0.05$ for 69/75 participants, one-way ANOVA; see examples in **Figure 2I**, **Figure 2—figure supplement 1**). The fitted models captured these effects adequately: the median Pearson's $r$ between model/participant mean RTs across stimulus position bins was 0.78 for horizontal position and 0.55 for vertical position (**Figure 2J**). While there was substantial variability across both participants and models in the particular position-dependent RT biases they exhibited, most participants/models responded more slowly when the stimuli were positioned close to the horizontal edges and, to a lesser extent, the vertical edges of the task window (**Figure 2—figure supplements 1 and 3**).

We also examined whether the VAMs captured two other commonly used behavioral metrics that take into account additional information from the RT distribution: RT delta plots and conditional accuracy functions (**De Jong et al., 1994**; **Ridderinkhof, 2002**; **Ulrich et al., 2015**; **White et al., 2011**; **van den Wildenberg et al., 2010**). The participant RT delta plots showed that the congruency effect increased with longer RTs (**Pratte, 2021**), a trend that was captured by the fitted VAMs (**Figure 2—figure supplement 4**). In contrast, we observed a mismatch between model and participant behavior for the conditional accuracy functions on incongruent trials (**Figure 2—figure supplement 4**). In particular, the participants tended to be less accurate on faster trials, while the models exhibited the opposite trend. This discrepancy may be explained by the lack of dynamic visual processing in the VAM, which has been proposed to account for the preponderance of errors on faster trials often observed in conflict tasks (**White et al., 2011**; **van den Wildenberg et al., 2010**). In the remainder of our analyses, we focus on the mean congruency effect, leaving a full accounting of such dynamic error patterns for future modeling work.

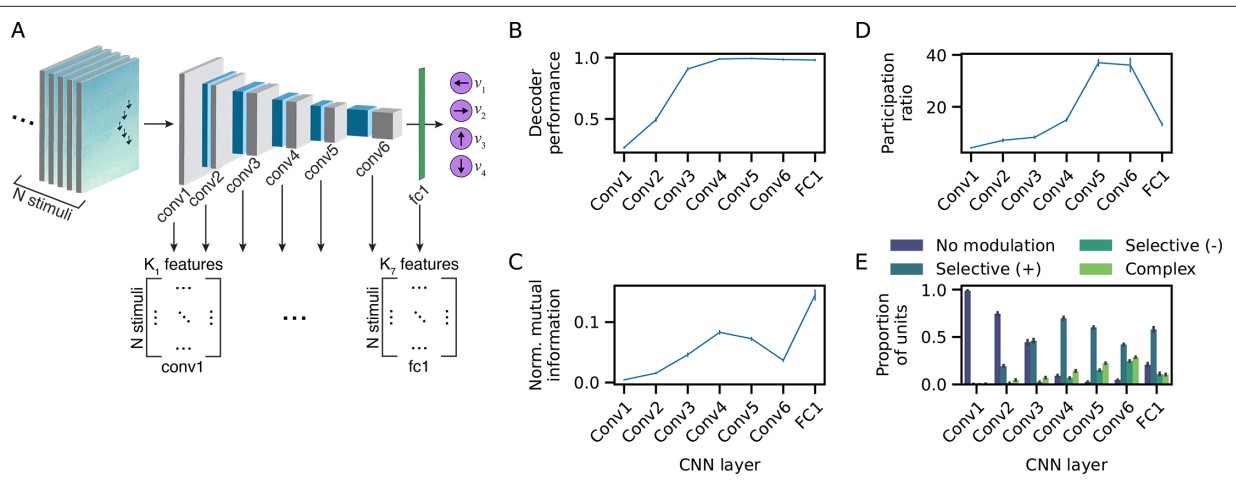

**Figure 3.** Neural representations of target direction. (**A**) Schematic of the convolutional neural network (CNN) activations extracted from each network layer. Each layer yields a $N \times K_l$ activation matrix, where $N$ is the number of stimuli and $K_l$ is the number of active units (i.e. feature dimensions) in layer $l$. (**B**) Decoding accuracy of stimulus target direction. (**C**) Normalized mutual information for target direction conveyed by single units, averaged across units. Mutual information was normalized by the entropy of the target direction distribution (possible range = [0, 1]). (**D**) Dimensionality of target representations as measured by the participation ratio of the target-centered activation covariance matrix. (**E**) Proportion of units exhibiting selectivity for target direction. Panels **B-E** show the average across $n = 75$ models; error bars correspond to bootstrap 95% confidence intervals.

The online version of this article includes the following figure supplement(s) for figure 3:

**Figure supplement 1.** Activity of all selective (+) units for one example model.

In summary, the VAM captured individual differences in RTs, accuracy, and congruency effects, and the dependence of RTs on stimulus layout and position. To lay the groundwork for understanding how the learned visual representations of the models relate to these behavioral effects, we sought to characterize the general properties of these representations that enable proficient execution of the task.

## Representations of task-relevant stimulus information

To perform LIM well, the models must learn representations that select the task-relevant information (target direction) and diminish the influence of irrelevant information (flanker direction, stimulus layout, and stimulus position). To investigate these representations, we presented the model with a holdout set of LIM stimuli (separate from the training stimuli) and analyzed the resulting unit activity in each CNN layer (*Hohman et al., 2020*; *Zeiler and Fergus, 2013*; *Figure 3A*, Methods). The activations from a given layer $l$ form an $N \times K_l$ matrix, where $N$ is the number of stimuli in the holdout set (5000) and $K_l$ is the number of active units in layer $l$ (a variable fraction of units in each layer did not respond to any stimuli and were excluded from the activation matrix).

To characterize the emergence of target selectivity, we adopted analysis techniques from the neuroscience literature that aim to determine what stimulus properties are coded by population-level neural activity, analogous to the high-dimensional CNN activations studied here. Specifically, we quantified how well the target direction could be decoded from the activation matrix of a given layer with a linear support vector machine (*Hung et al., 2005*; *Koren et al., 2020*; *Rust and Dicarlo, 2010*) (SVM; *Figure 3B*). Note that only incongruent trials were used in these analyses, since the classifier could use the flanker direction to classify the target on congruent trials, artificially inflating performance. Decoding performance on holdout data for target direction was near chance (27%) in the first network layer and increased to nearly perfect decoding ($\geq$ 97%) at layer 4, with a sharp increase between the second and third convolutional layers (*Figure 3B*). The increase in target decoding accuracy from shallower to deeper network layers is generally consistent with neural recording studies in mammals and other neural networks studies that document more accurate decoding of abstract variables (e.g. object or category identity) in higher visual/cortical regions and deeper neural network layers (*Brincat et al., 2018*; *Muratore et al., 2022*; *Tafazoli et al., 2017*).

We also investigated target selectivity at the single-unit level by quantifying the mutual information between each unit's activity and target direction (*Muratore et al., 2022*; *Tafazoli et al., 2017*;

*Figure 3C*). In contrast to the population-level decoding results, information for target direction exhibited a non-monotonic 'hunchback' profile across the convolutional layers, then increased sharply in the final fully-connected layer. The observation that single-unit information for target direction decreased between the fourth and final convolutional layers indicates that the units become progressively less selective for particular target directions. Since population-level decoding remained high in these layers, this suggests a transition from representing target direction with specialized 'target neurons' to a more distributed, ensemble-level code. Notably, a similar transition in coding properties takes place along the cortical hierarchy, with higher-order cortical regions exhibiting a reduction in units with 'pure' selectivity and a corresponding increase in units with mixed selectivity for multiple task or stimulus features (*Annis et al., 2021*; *Meister et al., 2013*; *Rigotti et al., 2013*). The high proportion of units with mixed selectivity results in a high-dimensional representation in which all task-relevant information can easily be extracted with simple linear decoders (*Cover, 1965*; *Pagan et al., 2013*).

Consistent with these ideas, we found that the dimensionality of target representations as measured by the participation ratio (Methods) increased sharply in the last two convolutional layers (Conv5–Conv6), paralleling the reduction in single-unit information for target direction in these layers (*Figure 3D*). The high-dimensional representation observed in these layers may enable the VAM to capture the rich stimulus-behavior dependencies present in the participant data. The reduction in dimensionality in the final fully-connected layer, and concomitant increase in single-unit information for target direction, may reflect a strong constraint to select the correct target direction imposed by the task. Notably, the hunchback-shaped profile we observed in the dimensionality of representations has also been observed along visual cortical regions of the rat ventral stream and in other neural network studies (*Ansuini et al., 2019*; *Muratore et al., 2022*).

To more explicitly characterize the degree of directional selectivity in each layer, we quantified the proportion of units that responded preferentially to one of the four target directions. We separated units according to the sign of modulation and degree of directional selectivity: 'selective (+)' and 'selective (-)' units were more (or less) active for one target direction relative to the other three; 'complex' units exhibited significant modulation by target direction without a clear directional preference (Methods).

The proportion of units that were selective for a particular direction increased steadily from the second to the fourth convolutional layer, with most units exhibiting positive modulation (*Figure 3E* and *Figure 3—figure supplement 1*). Between the fourth and final convolutional layers, the proportion of units with positive target direction modulation decreased, while the proportion of units with negative and complex modulation increased.

In summary, a strong representation for target direction emerged in the middle convolutional layers, and was initially supported by a simple low-dimensional code, with information for each target direction concentrated in separate populations of directionally-tuned units. In later convolutional layers, target direction was supported by a more distributed, complex, and high-dimensional code, with weaker directional selectivity at the single-unit level.

## Extraction and suppression of distracting stimulus information

How do the models' visual representations enable target selectivity for stimuli that vary along several irrelevant dimensions? The mammalian visual system solves the analogous problem of visual object recognition through representations that become increasingly invariant to different object positions, sizes, and lighting conditions (*DiCarlo et al., 2012*; *Rust and Dicarlo, 2010*). To determine whether the models learned representations that share these characteristics, we initially assessed whether the model representations for target direction are indeed invariant (or more generally, tolerant), to variation in flanker direction, stimulus layout, and horizontal/vertical position. Alternatively, the models could have learned a coding scheme in which different representations are responsible for selecting the target in each distracter context, e.g., with units that respond to the conjunction of particular target direction and layout combinations.

To assess the degree of representation tolerance, we quantified how well a linear SVM trained to classify target direction in one distracter context generalized to a different distracter context, where the distracter context is defined by a particular type of task-irrelevant information (*Bernardi et al., 2020*; *Rust and Dicarlo, 2010*). For example, to assess the degree of tolerance to flanker direction, we split trials into a training set where the flanker direction was fixed to a particular value (e.g. down),

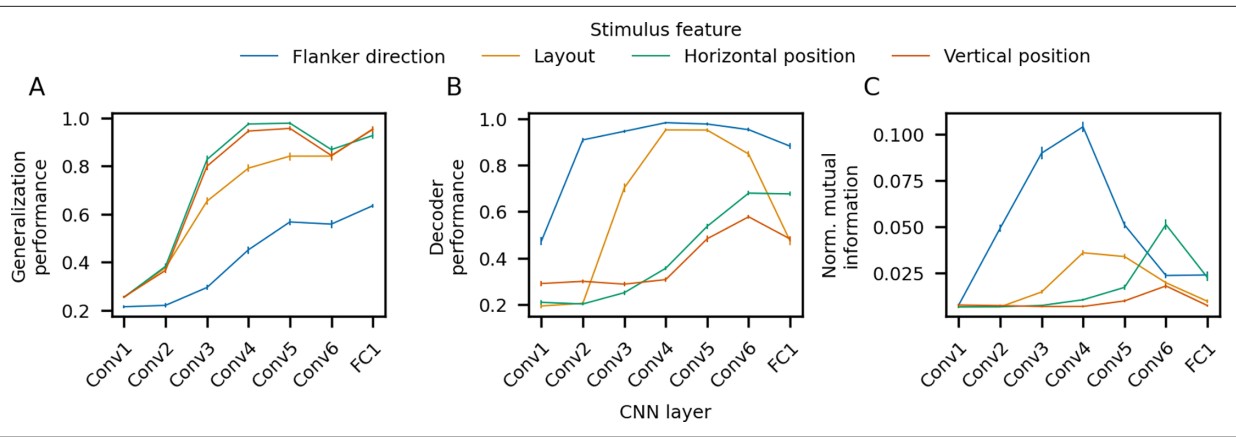

**Figure 4.** Suppression of task-irrelevant information and tolerance in task-relevant representations. (**A**) Decoding accuracy of stimulus target direction in a new distracter context (generalization performance). Context was defined by the values of a given stimulus feature (flanker direction, layout, horizontal/vertical position). (**B**) Decoding accuracy of irrelevant stimulus features. (**C**) Normalized mutual information for irrelevant stimulus features conveyed by single units, averaged across units. For each stimulus feature, the mutual information was normalized by the entropy of the stimulus feature distribution (possible range = [0, 1]). All panels show the average across *n* = 75 models; error bars correspond to bootstrap 95% confidence intervals.

and a generalization set with all other flanker directions (e.g. flanker direction = left/right/up). The performance of the classifier on the generalization set measures the extent to which the target representations are tolerant (or invariant) to variability in a given type of distracting information.

For each type of distracting information, we found that generalization performance was initially near chance (25%) and increased steadily through the network layers (*Figure 4A*). The lower generalization performance for flanker direction observed in the deepest network layers (~60%) can be attributed to the robust congruency effects exhibited by the models (*Figure 2D, E*): for the vast majority of incongruent error trials, the model chose the flanker direction, implying that flanker direction has a strong impact on model representations.

The tolerance of target direction representations to variability in irrelevant stimulus features suggests that the irrelevant information was progressively suppressed from shallower to deeper network layers. To examine this explicitly, we quantified how well each irrelevant stimulus feature (flanker direction, stimulus layout, horizontal/vertical position) could be decoded from the activity in each layer using the same SVM classifier methodology as we did for target direction decoding. We found that decoding accuracy for flanker direction and stimulus layout exhibited a hunchback profile in which decoding performance started low in early layers, increased in intermediate layers, and decreased in later layers (*Figure 4B*). The decoding accuracy for horizontal and vertical position followed a similar but shifted pattern: there was a steady increase in accuracy until the second-to-last layer, followed by a slight drop in the last layer. Partial (rather than complete) suppression of irrelevant stimulus features is expected given that these features all impact behavior (*Figure 2*). Note that the increase in receptive field size that occurs from shallower to deeper layers is necessary but not *sufficient* for accurate decoding of the irrelevant stimulus features, and does not explain the reduction in decoding accuracy in later layers.

We also examined suppression at the single-unit level by quantifying the mutual information between each unit's activations and the values of each task-irrelevant stimulus feature (*Tafazoli et al., 2017*; *Figure 4C*). The mutual information for a given stimulus feature was normalized by the entropy of the feature distribution to facilitate comparisons between the different stimulus features (*Muratore et al., 2022*). In agreement with the population-level decoding analyses, information for the irrelevant stimulus features exhibited a pronounced hunchback profile in the progression from shallower to deeper network layers. It is noteworthy that the suppression of irrelevant information is more pronounced at the single-unit level in that decoding accuracy remains relatively high in later layers, particularly for flanker direction. This parallels the findings discussed above for target direction, and again suggests a transition from a simple code with populations of units that are selective for particular stimulus features to a more distributed, ensemble-level code.

## Orthogonality of task-relevant and irrelevant information predicts behavior

A noteworthy feature of visual attention is that the selectivity and tolerance identified above is not absolute: irrelevant information cannot be filtered out completely, as illustrated by the congruency effects observed in the flanker task and related conflict tasks. A common framework for understanding these behavioral phenomena posits that task-relevant and irrelevant information compete for control over response execution, and that congruency effects arise from incomplete suppression of irrelevant information (*Ulrich et al., 2015*; *van den Wildenberg et al., 2010*). Motivated by these theories, we investigated whether the degree of suppression of irrelevant information in the trained models was correlated with congruency effects across the models we analyzed.

To this end, we operationalized suppression with two metrics of the model representations that we investigated above: the accuracy of decoders trained to classify flanker direction from the model representations and the mutual information for flanker direction conveyed by single units. We expected to observe a positive correlation between both of these metrics and both RT and accuracy congruency effects: higher decoder accuracy or mutual information for flanker direction corresponds to less suppression and, therefore, higher congruency effects. However, we did not observe a significant positive correlation between either decoding accuracy or mutual information and RT or accuracy congruency effects in any of the model layers, with the exception of a single significant positive correlation between flanker mutual information and accuracy congruency effects in layer Conv3 (*Figure 5—figure supplement 1*).

Given this somewhat surprising negative result, we were motivated to consider an alternative account of congruency effects, one that takes into account the relative geometry of the task-irrelevant (flanker) *and* task-relevant (target) information. In particular, we considered the possibility

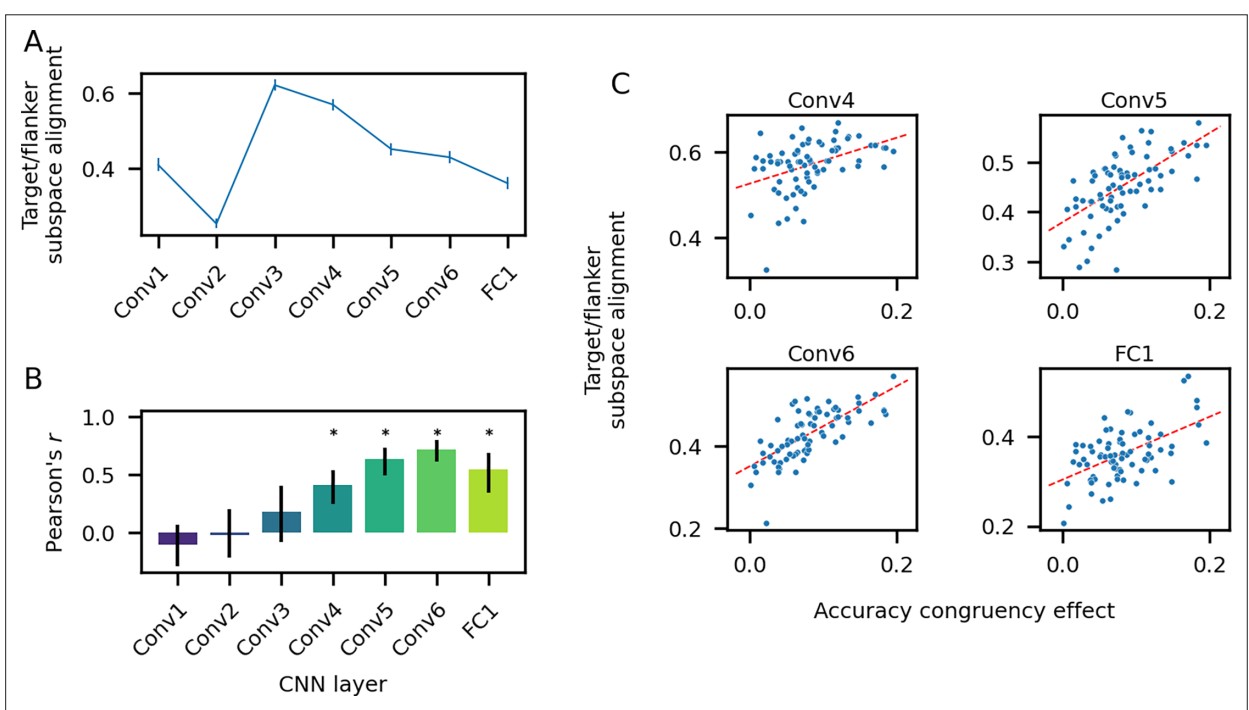

**Figure 5.** Orthogonality of target/flanker subspaces predicts accuracy congruency effects. (**A**) Target/flanker subspace alignment averaged across models. (**B**) Pearson's correlation coefficient between target/flanker subspace alignment and accuracy congruency effect calculated across models. (**C**) Target/flanker subspace alignment vs. accuracy congruency effect for layers Conv4–FC1. Each point corresponds to one model; the red line is the linear best fit. For all panels, *n* = 75 models. Error bars in panels **A–B** correspond to bootstrap 95% confidence intervals. Asterisks in panel **B** indicate a significant Pearson's *r* (adjusted *p*-value<0.05, permutation test with *n* = 1000 shuffles, Bonferroni correction for seven comparisons).

The online version of this article includes the following figure supplement(s) for figure 5:

**Figure supplement 1.** Absence of correlation between flanker suppression metrics and congruency effects.

**Figure supplement 2.** Absence of correlation between target/flanker subspace alignment and response time (RT) congruency effect.

that task-relevant and irrelevant information could be orthogonalized in the high-dimensional space of neural activity, such that task-relevant information is shielded from distracter interference. The general idea that neural representations for different types of information can be orthogonalized to prevent interference has received support from a number of neural recording studies (*Flesch et al., 2022*; *Kaufman et al., 2014*; *Libby and Buschman, 2021*; *Panichello and Buschman, 2021*; *Ritz and Shenhav, 2024*).

To examine whether target and flanker representations are orthogonalized, we first defined target and flanker subspaces from the target direction and flanker direction classifiers used in the decoding analyses described above (Methods). The target direction classifier for a given network layer implicitly defines four decoding vectors, one for each target direction (*Bernardi et al., 2020*; *Libby and Buschman, 2021*). We define the subspace of the $K_l$-dimensional feature space spanned by these four vectors as the target subspace; the flanker subspace is defined analogously.

To measure the orthogonality between target and flanker subspaces, we calculated the average of the cosine of the principal angles between the target and flanker subspaces, a metric we refer to as subspace alignment (Methods). Principal angles generalizes the idea of angles between lines or planes to arbitrary dimensions (*Jordan, 1873*). The subspace alignment metric has a simple interpretation: it is equal to one if the subspaces are completely parallel and zero if they are completely orthogonal.

We first characterized how target/flanker subspace alignment develops across the layers of the trained models (*Figure 5A*). Given that target direction decoding is poor in the first two layers (<50%; *Figure 3B*), the decoding vectors used to define the target subspace are not particularly meaningful, and we do not attempt to interpret the subspace alignment metric in these layers. Beginning at the third convolutional layer, when decoding accuracy for both targets and flankers is high (>90%), we found that target and flanker subspaces are well-aligned. We interpret the high alignment as evidence that the model has learned a common representation for direction that is shared for both targets and flankers. In later layers, we found that target and flanker representations become increasingly orthogonal, consistent with the view that the processing in later layers acts to reduce interference between task-relevant and irrelevant information.

If orthogonalizing target and flanker representations reduces interference from the irrelevant (flanker) information, we should observe a positive correlation between subspace alignment and congruency effects across models, since greater alignment results in more interference. Consistent with this idea, we observed a significant positive correlation between subspace alignment and accuracy congruency effects across models in each layer beginning with the fourth convolutional layer (*Figure 5B andC*; adjusted *p*-value<0.05 for layers 3–7, permutation test, Bonferroni correction for seven comparisons). In contrast, we did not observe a significant correlation between target/flanker subspace alignment and RT congruency effects in any network layer, suggesting a mechanistic dissociation between RT and accuracy congruency effects (*Figure 5—figure supplement 2*).

## Representation geometry of task-optimized models

Researchers who use neural network models to study neural representations typically optimize the model to perform a task, rather than fit the model to behavioral data from the task as we do here *Mante et al., 2013*; *Wang et al., 2018*; *Yamins et al., 2014*; *Yang et al., 2019*; *Dezfouli et al., 2019*; *Jaffe et al., 2023*; *Sussillo et al., 2015*. This raises the possibility that these two training paradigms induce different representations in the models. To investigate this, we trained CNNs to perform LIM (i.e. output the direction of the target) by minimizing the standard cross-entropy loss function used in image classification tasks, where the training labels are given by the true direction of the target bird in each stimulus. The task-optimized CNNs were identical to those used in the VAMs, except that the outputs of the last layer were converted to softmax-scored probabilities for each direction rather than drift rates. Otherwise, all aspects of the optimization algorithm, CNN architecture, initialization, and training data for the task-optimized models were the same as those used to train the VAMs. We trained one task-optimized model for each VAM using stimulus data from the same participants (*n*=75 task-optimized models).

We first compared the behavioral outputs of the task-optimized models and the VAMs. We found that the task-optimized models did not exhibit an accuracy congruency effect (*Figure 6A*). Thus, simply training the model to perform the task is not sufficient to reproduce a behavioral phenomenon widely-observed in conflict tasks. This challenges a core (but often implicit) assumption of the

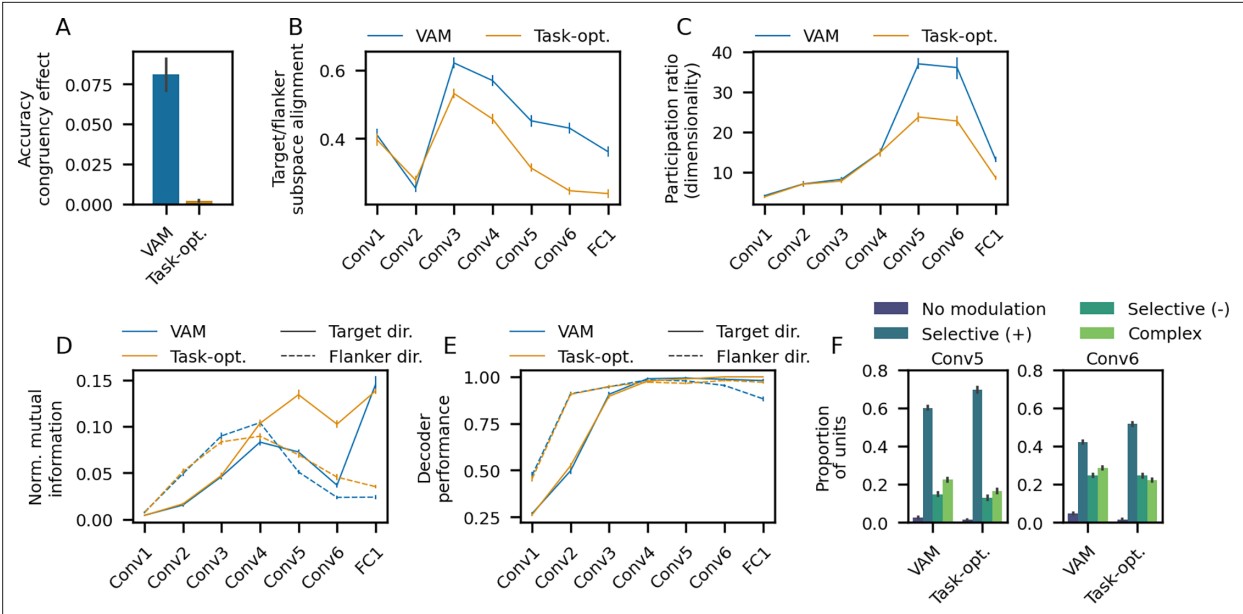

**Figure 6.** Comparison of visual accumulator models (VAMs) and task-optimized models. (**A**) Accuracy congruency effect. (**B**) Target/flanker subspace alignment. (**C**) Dimensionality of target representations, as measured by the participation ratio of the target-centered activation covariance matrix. (**D**) Normalized mutual information for target/flanker direction conveyed by single units, averaged across units. Mutual information was normalized by the entropy of the target/flanker direction distribution (possible range = [0, 1]). (**E**) Decoding accuracy of target/flanker direction. (**F**) Proportion of units exhibiting selectivity for target direction in layers Conv5–Conv6. All panels show the average across *n* = 75 task-optimized models and *n* = 75 VAMs; error bars correspond to bootstrap 95% confidence intervals. The VAM data shown in panels **A–F** is the same as that shown in *Figures 2E, 5A and 3B-E* and *Figure 4B, C*, respectively.

The online version of this article includes the following figure supplement(s) for figure 6:

**Figure supplement 1.** Additional analysis of visual accumulator models (VAMs) and task-optimized models.

task-optimized training paradigm, namely that training a model to do a task well will result in model representations that are similar to those employed by humans. Indeed, for a number of visual tasks, the representations and behavior of task-optimized CNNs has been observed to differ considerably from those of humans (*Eckstein et al., 2017*; *Jacobs and Bates, 2019*, *Jha et al., 2023*; *Rajalingham et al., 2018*; *Sanders and Nosofsky, 2020*).

Since the task-optimized models do not generate RTs, it is not possible to directly measure RT congruency effects in these models without making additional assumptions about how the CNN's classification decisions relate to RTs. However, as a coarse proxy for RT, we can examine the confidence of the CNN's decisions, defined as the softmax-scored logit (probability) of the most probable direction in the final CNN layer. This choice of RT proxy is motivated by some prior studies that have combined CNNs with EAMs (*Annis et al., 2021*; *Holmes et al., 2020*; *Trueblood et al., 2021*). These studies explicitly or implicitly derive a measure of decision confidence from the activity of the last CNN layer. The confidence measure is then mapped to the EAM drift rates, such that greater decision confidence generally corresponds to higher drift rates (and therefore shorter RTs).

We calculated the average confidence of each task-optimized CNN separately for congruent vs. incongruent trials. On average, the task-optimized models showed higher confidence on congruent vs. incongruent trials (*W*=21.0, p<1e-3, Wilcoxon signed-rank test; Cohen's *d* = 0.99; *n* = 75 models). Therefore, these analyses provide some evidence that task-optimized CNNs have the capacity to exhibit congruency effects, though an explicit comparison of the magnitude of these effects with human data requires additional modeling assumptions (e.g. fitting a separate EAM).

Above, we showed that VAMs with greater orthogonalization of target and flanker information exhibit smaller accuracy congruency effects (*Figure 5B*), providing evidence that the relative geometry of task-relevant and irrelevant representations is a critical determinant of the degree of flanker interference at the behavioral level. Given that the task-optimized models do not exhibit an accuracy congruency effect, we expect that these models would exhibit a higher degree of orthogonalization

of target and flanker information. Consistent with this idea, we found that the task-optimized models had lower target/flanker subspace alignment (i.e. higher orthogonalization) for all network layers beginning with the third convolutional layer (*Figure 6B*).

A direct consequence of the training paradigms used to train the VAMs is that these models are encouraged to capture dependencies between the stimulus features and behavior (RTs and accuracy), while the task-optimized models are not. As a result, the VAMs may learn more complex representations of the stimuli, since a variety of stimulus features—layout, stimulus position, flanker direction—influence behavior (*Figure 2*). To investigate this possibility, we compared the dimensionality of target representations between the VAMs and task-optimized models using the participation ratio metric discussed above.

We found that the dimensionality of the VAM and task-optimized model representations was nearly identical for the first four convolutional layers (*Figure 6C*). In contrast, for the final two convolutional layers, the VAMs exhibited a substantially more pronounced expansion of dimensionality than the task-optimized models. In the final fully-connected layer, dimensionality decreased sharply for both types of models, and was somewhat higher for the VAMs.

The increased dimensionality of VAMs' target represenations in later network layers is consistent with the view that these models must learn more complex representations of the stimuli in order to successfully capture stimulus-behavior dependencies. It is also noteworthy that the most striking difference between the dimensionality of the VAMs and task-optimized models occurs during the latter part (layers Conv5–Conv6) of the expansion phase of the hunchback-shaped dimensionality profile discussed above and observed in prior work (*Ansuini et al., 2019*; *Mante et al., 2013*). In these layers, single-unit information for target and flanker direction—the primary task features—decreases, while population-level decoding of these features remains high (*Figures 3B–C , and 4B–C*). As discussed above, this dissociation implies a transition from a simple representation of target/flanker direction with separate populations of directionally-tuned units to a more complex and distributed code.

To determine whether the task-optimized models exhibited this change in coding properties, we quantified the single-unit information and population-level decoding accuracy for target/flanker direction in these models. Relative to the VAMs, the task-optimized models had substantially higher single-unit information for both target and flanker direction in layers Conv5–Conv6 (*Figure 6D*). The task-optimized models also showed marginally more or roughly equivalent single-unit information for stimulus position in these layers relative to the VAMs, but had less single-unit information for stimulus layout (*Figure 6—figure supplement 1*).

In contrast, population-level decoding accuracy for target/flanker direction and stimulus position was similar between the task-optimized models and VAMs in layers Conv5–Conv6, though decoding accuracy for stimulus layout was notably lower for the task-optimized models (*Figure 6E* and *Figure 6—figure supplement 1*). These results suggest that the task-optimized models maintained a simpler code for target/flanker direction in the later convolutional layers relative to the VAMs, primarily relying on separate populations of directionally-tuned units. Consistent with this idea, the task-optimized models had a higher proportion of simple selective (+) directionally-tuned units and a lower proportion of complex units in layers Conv5–6 relative to the VAMs (*Figure 6F*).

## Discussion

The dominant models of decision-making, while providing a good quantitative description of psychophysical data, do not incorporate biologically plausible models of the perceptual processes that are essential for many behaviors (*Evans and Wagenmakers, 2020*). On the other hand, neural network models of vision, while capturing core properties of the primate visual system, do not account for many results from behavioral experiments (*Baker et al., 2018*; *Bowers et al., 2022*; *Fel et al., 2022*; *Linsley et al., 2017*; *Malhotra et al., 2022*). The VAM addresses both of these limitations by integrating neural network models for visual processing with traditional decision-making models in a unified probabilistic framework that can be fitted to visual stimuli and RT data. Leveraging large-scale data from a task with rich visual stimuli, we demonstrate that our framework captures complex dependencies between stimulus features and behavior at the level of individual participants. We also illustrate how congruency effects—a core behavioral phenomenon observed in conflict tasks—can be explained in terms of the visual representations of the model. Finally, we document several key

differences between the representations learned by models fitted to human behavioral data (the VAMs) and those learned by models trained only to do the task.

## Processing phases underlying the transformation of sensory information

To perform the task and capture the behavioral data, the VAMs learned representations that—like the mammalian cortex—extract task-relevant information from raw visual inputs. Our analyses of these representations revealed several discrete processing phases that are fruitfully discussed in relation to the changes in target representation dimensionality we observed. Across the layers of the VAM's CNN, target dimensionality exhibited a prominent hunchback shape profile, corresponding to an initial protracted phase of dimensionality expansion followed by abrupt dimensionality compression in the final network layers. An analogous expansion and subsequent compression of object representation dimensionality has been documented in CNNs trained to classify images and along the rat visual-cortical hierarchy (*Ansuini et al., 2019*; *Muratore et al., 2022*), with some notable differences from our work that we highlight below.

We speculate that the initial phase of dimensionality expansion can be explained in part by the pruning of low-level stimulus features (e.g. contrast and luminosity) that are correlated across stimuli in the dataset (*Ansuini et al., 2019*). These correlations are particularly strong in our dataset since we used the same background image for each stimulus, resulting in low-dimensional activity in the initial network layers. Conceivably, the initial increase in target representation dimensionality (i.e.layers Conv1–Conv4) results from the removal of these correlations, analogous to a whitening transformation (*Ansuini et al., 2019*).

In the early and middle convolutional layers (Conv1–Conv4), we found that the population-level decoding and single-unit information for both task-relevant (target direction) and distracting information (flanker direction, stimulus layout, and position) increased, occurring in parallel with the subtle expansion of dimensionality we observed in these layers. These trends are broadly consistent with observations that the mammalian visual cortex encodes increasingly abstract stimulus properties (e.g. object identity) in downstream brain regions (*Güçlü and van Gerven, 2015*; *Hung et al., 2005*; *Rust and Dicarlo, 2010*). The fact that information for distracting stimulus features increased in these layers is especially noteworthy, given that this information—the flanker direction, most prominently—impairs performance on the task (*Eriksen and Eriksen, 1974*; *Stoffels and van der Molen, 1988*). While it is tempting to speculate that the VAMs learned to extract distracter information because they were fitted to behavioral data, where the impairments in task performance manifest, the task-optimized models also exhibited increased distracter information in these layers. This suggests an alternative explanation in which the model first extracts more granular, superordinate stimulus representations that group together stimulus features with similar statistics. For example, the initial increase in encoding of target and flanker direction may reflect the representation of a unitary 'directional signal' that facilitates the eventual separation of target and flanker direction representations that occurs in later layers. This idea is consistent with our observation that target/flanker subspaces are highly aligned in the middle convolutional layers and become progressively more orthogonal in later network layers.

In the last convolutional layers (Conv5–Conv6), we observed a large expansion in target representation dimensionality, which was notably less pronounced in the task-optimized models compared to the VAMs. We speculate that the increase in dimensionality may be partly due to an increase in mixed selectivity for multiple stimulus features at the single-unit level, similar to the single-unit coding properties observed in primate PFC (*Rigotti et al., 2013*). In general agreement with this idea, we observed a reduction in the proportion of units with selectivity for particular target directions in these layers, and a concomitant increase in units that were modulated by multiple target directions. Relative to the VAMs, the task-optimized models showed a higher proportion of directionally-tuned units in these layers. A notable advantage of high-dimensional neural codes composed of units with mixed selectivity is that many different input-output relationships can be implemented with simple decoders (*Cover, 1965*; *Pagan et al., 2013*; *Rigotti et al., 2013*). Thus, the higher dimensionality and more complex target direction coding observed in the VAMs relative to the task-optimized models may reflect the fact that the VAMs are trained to capture a potentially large number of dependencies between stimulus attributes and behavior, while the task-optimized models need only extract the target direction. Given the impressive capacity of primates to learn complex tasks and arbitrary

stimulus behavior relationships, and the abundance of high-dimensional representations across the cortex, models trained to capture the richness of stimulus-behavior dependencies—such as the VAM—may result in better models of cortical processing than optimizing models to perform tasks.

## Congruency effects emerge from the relative geometry of task-relevant vs. distracting sensory representations

One of the key strengths of our modeling framework is that neural representations of stimulus features can be directly related to behavioral outputs. We focused in particular on congruency effects since they are ubiquitously observed in conflict tasks and highlight an inherent limitation of selective attention, namely that humans cannot completely filter out distracting information. Congruency effects are often interpreted in the context of a 'dual-process' model in which automatic or impulsive processing arising from distracting stimulus information and more controlled or deliberate processing of task-relevant information compete for control over behavior (*Ridderinkhof, 2022*; *Ulrich et al., 2015*; *van den Wildenberg et al., 2010*). According to this model, congruency effects result from the incomplete suppression of incorrect response activation in the automatic pathway. Building on this framework, a variety of models have been proposed that successfully capture congruency effects and other behavioral phenomena observed in conflict tasks (*Cohen et al., 1992*; *Ulrich et al., 2015*; *White et al., 2011*).

Our work differs from these prior modeling efforts in two key ways. First, we did not attempt to 'build in' a predetermined implementation of congruency effects. Rather, we fitted a relatively unstructured neural network model to human flanker task data and examined how congruency effects emerged (*Jaffe et al., 2023*). Second, we explicitly modeled how task-relevant (and task-irrelevant) representations are extracted from raw visual stimuli with a biologically-plausible model of the mammalian visual system (a CNN). These features of the VAM allowed us to explore a space of possible explanations for congruency effects that are not readily investigated with other models of conflict tasks. Each CNN layer is composed of many simple neuron-like processing units, enabling a population-level description of task-relevant and irrelevant representations. Prior connectionist models of conflict tasks are also implemented as networks of interacting processing units (*Cohen et al., 1992*), but are much reduced in scale relative to the VAM, precluding a characterization of stimulus feature representations in terms of their high-dimensional geometry as we pursue here. Another relevant attribute of CNNs is that the successive convolution and pooling operations in each layer extract increasingly abstract, task-related representations from raw visual inputs, a hallmark of the mammalian ventral visual system. This enabled a rich characterization of the intermediate visual representations that sequentially transform raw visual stimuli to behavioral outputs.

Leveraging these features of the VAM framework, we investigated potential explanations for congruency effects in terms of the population-level activity and representation geometry in each network layer. Motivated by the dual-process model of conflict tasks mentioned above, we initially sought evidence for the suppression of task-irrelevant information, as operationalized by population-level decoding accuracy and single-unit information for flanker direction. While we did find evidence of greater suppression in later vs. middle convolutional layers for both of these metrics, variability in the magnitude of this suppression across models was not related to variability in congruency effects.

Instead, we found that the relative geometry of target and flanker representations was a critical determinant of congruency effects. In particular, models with smaller accuracy congruency effects had more orthogonal (or less aligned) representations of targets and flankers in later network layers, proximal to behavioral outputs. This finding is consistent with the idea that orthogonalization of task-relevant/irrelevant representations shields the task-relevant information from distracters, and parallels findings from neural recording studies in both humans and animals that representations for different sources of information can be orthogonalized in order to prevent interference (*Flesch et al., 2022*; *Kaufman et al., 2014*; *Libby and Buschman, 2021*; *Panichello and Buschman, 2021*; *Ritz and Shenhav, 2024*; *Xie et al., 2022*).

To elaborate on this idea, in models with more aligned (less orthogonal) target/flanker representations in intermediate layers, the task-relevant (target) subspace is effectively 'contaminated' by task-irrelevant (flanker) information. In the absence of any corrective process, the task-relevant subspace propagates a mixture of both correct (target) and incorrect (flanker) direction signals forward through the network. At the final readout layer of the network, assuming that the target subspace is

well-aligned with the readout weights (*Papyan et al., 2020*), flanker signals within the target subspace then contribute to the drift rate for the (incorrect) flanker direction. This increases the probability of an error, such that models with more aligned target/flanker representations exhibit larger accuracy congruency effects. A corollary of this proposition is that the absolute amount of flanker information in the network—equivalently, the degree to which flanker information has been suppressed—is not necessarily predictive of congruency effects. This may account for our observations that the measures of suppression we considered are not correlated with accuracy congruency effects across models, and that representations for flanker direction do not appear to be strongly suppressed even in later network layers, as evidenced by high decoding accuracy for flanker direction in both the VAMs and task-optimized models.

### The role of dynamics in conflict tasks

One apparent limitation of the VAM as presented here is that it does not have visual processing dynamics, which seem to be required to explain some observations from the flanker task and related conflict tasks. For example, the RTs on incongruent error trials are typically faster than error RTs for congruent trials and RTs for correct trials *van den Wildenberg et al., 2010*, an effect that we confirm is also present in the flanker task variant studied here. This observation can be explained by a 'shrinking attention spotlight' in which response activation from flankers starts high and diminishes over time, resulting in a higher proportion of errors for faster RTs *White et al., 2011*. We speculate that our models were unable to capture these particular error patterns because the visual processing module (CNN) we used does not have any dynamics (e.g. recurrence) that could instantiate such time-varying attention and resultant time-varying drift rates. However, it is not difficult to imagine how the orthogonalization mechanism described above, which explains variability in accuracy congruency effects *across* individuals, could act in concert with other dynamic processes that explain variability in congruency effects *within* individuals (e.g. as a function of RT). In general, any process that dynamically gates the influence of irrelevant sensory information on behavioral outputs could accomplish this, for example, ramping inhibition of incorrect response activation (*van den Wildenberg et al., 2010*), a shrinking attention spotlight (*White et al., 2011*), or dynamics in neural population-level geometry (*Kaufman et al., 2014*). To pursue these ideas, future work may aim to incorporate dynamics into the visual component and decision component of the VAM with recurrent CNNs (*Goetschalckx et al., 2023*; *Nayebi et al., 2018*) and the task-DyVA model (*Jaffe et al., 2023*), respectively.

### Conclusion

The VAM is a probabilistic model of psychophysical data that captures how raw sensory inputs are transformed into the abstract representations that guide decisions. Raw (pixel-space) visual stimuli are processed by a biologically-plausible neural network model of vision that outputs the parameters of a traditional decision-making model. Each VAM is fitted to data from a single participant, a feature that allowed us to study how individual differences in behavior emerge from differences in the 'brains' of the models. To this end, we found that models with smaller congruency effects had more orthogonal representations for task-relevant and irrelevant information. While we chose to use a CNN to model visual processing, we note that the VAM is not limited to this choice: other sensory encoding models, such as those based on transformer architectures (*Dosovitskiy et al., 2021*), can be readily swapped in to replace the CNN with minimal changes to the underlying VAM implementation. Similarly, the LBA decision-making model we employed is easily replaced with other decision-making models that have a closed-form or easily approximated likelihood, such as the diffusion-decision model and leaky competing accumulator model (*Lo and Ip, 2021*; *Navarro and Fuss, 2009*; *Ratcliff and Rouder, 1998*; *Usher and McClelland, 2001*). In this way, the VAM provides a general probabilistic framework for jointly fitting interpretable decision-making models and expressive neural network architectures of sensory processing with psychophysical data.

## Methods

### Datasets

We used deidentified Lost in Migration gameplay data from 75 Lumosity users (participants) to train the models included in this paper. The mean (SD) age of this sample at the time of signup was 56.4

(18.6) y; 46.7% identified as female, 48.0% identified as male, and 5.3% did not report their gender. All analysis and modeling was done retrospectively on preexisting Lumosity data that were collected during the normal use of the Lumosity program (at home). All participants consented to the use and disclosure of their deidentified Lumosity data for any purpose as laid out in the Lumosity Privacy Policy (https://www.lumosity.com/en/legal/privacy_policy/). No statistical methods were used to predetermine sample sizes, though our sample sizes are comparable to or larger than those used in related modeling work (*Brown and Heathcote, 2008*; *White et al., 2011*).

The included participants were selected from a larger pool that met certain inclusion criteria. The selection from this larger pool was done at random, with the following exception: we included all participants under the age of 40 (n=19) to ensure adequate representation of younger participants. The criteria used to define the larger participant pool were as follows: we required that participants had signed up as Lumosity users between June 28, 2015 and June 30, 2020 (inclusive); that they were between the ages of 18 and 89 at the time of signup (inclusive); that their country of origin was Australia, Canada, New Zealand, or the United States; that their preferred language was English; and that they were not employees of Lumos Labs, Inc Accounts created for research purposes were also excluded. We also required that all of a given participant's Lost in Migration gameplays were done on the web (as opposed to mobile) platform. Finally, we required that participants had at least 200 Lost in Migration gameplays and at least 25,000 trials starting with their 50th gameplay. One additional participant with near chance accuracy (33%) on Lost in Migration was excluded from the larger participant pool.

Trials with very short and very long RTs were excluded and did not count toward the 25,000 trial minimum. Specifically, we excluded trials less than or equal to 250 ms and trials classified as outliers from a criterion based on the median absolute deviation from the median (the MAD): trials with an absolute deviation from the median RT of more than 10 times a given participant's MAD were excluded.

The final datasets from each participant used for model training consist of the first 25,000 non-outlier trials starting with their 50th gameplay. Data from the first 49 gameplays were excluded to reduce learning-related variability. Approximately 50% of trials were congruent (vs. incongruent).

## Modeling framework

### Linear ballistic accumulator (LBA) model

The decision-making component of our modeling framework is the LBA model (*Kumbhar et al., 2020*), tailored to Lost in Migration. Evidence for each of the four possible response directions is accumulated linearly and independently at a response-specific drift rate $d_k$. Commitment to a decision occurs when one of the accumulators reaches a fixed threshold parameter $b$. The RT on a given trial is the duration of this evidence accumulation process plus a constant non-decision time parameter $t_0$ that includes both sensory processing and motor execution. Response variability comes from two sources. First, on each trial, the drift rate $d_k$ for each response $k \in \{1, 2, 3, 4\}$ is sampled independently from a Gaussian distribution with mean $v_k$ and a common SD $s$. We fix $s$ to 1 for all models to ensure that the LBA parameters are identifiable (*Dao et al., 2024*; *Gunawan et al., 2020*). Second, the initial evidence for each accumulator is sampled independently from a uniform distribution on the interval $[0, A]$, where $A$ is a model parameter.

### Visual accumulator model (VAM): Overview

The VAM is a generalization of the LBA model in which the drift rate means $v^{(i)} = \{v_1^{(i)}, v_2^{(i)}, v_3^{(i)}, v_4^{(i)}\}$ on trial $i$ depend on the stimuli $s^{(i)}$ through a CNN. For a dataset with $N$ trials, the task data are $\mathbf{x} = \{$response times $\mathbf{t}$, choices $\mathbf{c}$, stimuli $\mathbf{s}\}$, where $\mathbf{t} = \{t^{(i)}\}_{i=1}^N$, $\mathbf{c} = \{c^{(i)}\}_{i=1}^N$, and $\mathbf{s} = \{s^{(i)}\}_{i=1}^N$. We adopt a Bayesian framework and model the joint density of the task data $\mathbf{x}$ and LBA parameters $\boldsymbol{\theta} = \{b, A, t_0\}$ as:

$$p(\mathbf{x}, \boldsymbol{\theta}) = \prod_{i=1}^N p(t^{(i)}, c^{(i)} | v^{(i)}, b, A, t_0) p(b, A, t_0), \tag{1}$$

$$v^{(i)} = \text{CNN}_\zeta(s^{(i)}), \tag{2}$$

where the drift rate means $v^{(i)}$ depend on a CNN with parameters $\zeta$ (described below). The factor on the left of *Equation (11)* is the likelihood of the LBA model, which has a closed-form solution (*Kumbhar et al., 2020*). The factor on the right is the prior distribution over the LBA parameters, specified as a standard multivariate Gaussian $p(\boldsymbol{\theta}) \sim \mathcal{N}(\mathbf{0}, \mathbf{I})$. We express the joint density more compactly as:

$$p(\mathbf{x}, \boldsymbol{\theta}; \zeta) = p(\mathbf{t}, \mathbf{c}|\text{CNN}_\zeta(s), b, A, t_0)p(b, A, t_0). \tag{3}$$

To fit the VAM, i.e., learn the posterior distribution over the LBA parameters $p(\boldsymbol{\theta}|\mathbf{x})$ and simultaneously optimize the CNN parameters $\zeta$, we apply automatic differentiation variational inference (ADVI) (*Kucukelbir et al., 2017*). Rather than attempting to sample from the true posterior directly as in Markov chain Monte Carlo (MCMC) methods, variational inference introduces an approximate posterior density $q(\boldsymbol{\theta}; \phi)$ and minimizes the Kullback-Leibler (KL) divergence from $p(\boldsymbol{\theta}|\mathbf{x})$ to $q(\boldsymbol{\theta}; \phi)$ by optimizing $\phi$. Here, we specify the approximate posterior $q(\boldsymbol{\theta}; \phi)$ as a multivariate Gaussian with mean $\mu$ and unconstrained covariance matrix $\boldsymbol{\Sigma}$ (thus $\phi = \{\mu, \boldsymbol{\Sigma}\}$).

We use variational inference since it scales well to large datasets and can handle complicated models (such as the VAM), in contrast to MCMC methods. Leveraging automatic differentiation software (JAX *Bradbury et al., 2018*), we automate the calculation of the derivatives of the variational objective (described below) with respect to both the CNN parameters $\zeta$ and variational parameters $\phi$.

## Variable transformations

Note that the LBA parameters $\boldsymbol{\theta} = \{b, A, t_0\}$ are restricted to be non-negative, while $\mu \in \mathbb{R}^3$ and $\boldsymbol{\Sigma}$ are restricted to be positive semidefinite. However, to optimize the variational objective (defined below), we require that $\phi$ and $\boldsymbol{\theta}$ have the same support (*Hohman et al., 2020*). To achieve this, we transform $\phi$ and $\boldsymbol{\theta}$ to both have support on the real line. Specifically, we reparameterize $\boldsymbol{\Sigma}$ using the Cholesky decomposition as $\boldsymbol{\Sigma} = \mathbf{L}\mathbf{L}^\mathsf{T}$, where $\mathbf{L}$ is a lower-triangular matrix with entries in $\mathbb{R}$ (so that now $\phi = \{\mu, \mathbf{L}\}$). For the LBA parameters, in addition to mapping them to the real line, we must also enforce the constraint that the threshold $b$ is always greater than the parameter $A$ controlling the range of the initial evidence distribution. We enforce this by defining $\tilde{b} = b - A$ and taking the log of $\tilde{b}$, $A$, and $t_0$ to map them to the real line (*Annis et al., 2021*). We define the transformed parameters as $\boldsymbol{\theta}^* = \{b^*, A^*, t_0^*\} = \{\log(b - A), \log A, \log t_0\}$, and we define the transformation that maps $\boldsymbol{\theta}$ to $\boldsymbol{\theta}^*$ as $T$.

## VAM objective function

To fit the VAM, we maximize the evidence lower bound (ELBO):

$$\mathcal{L}(\phi, \zeta) = \mathbb{E}_{q_\phi(\boldsymbol{\theta})}[\log p(\mathbf{x}, \boldsymbol{\theta}; \zeta) - \log q(\boldsymbol{\theta}; \phi)].$$

This is the standard ELBO optimized in variational inference, with an additional dependence on the CNN parameters $\zeta$. The ELBO is a lower bound on the marginal likelihood of the data $p(\mathbf{x})$, and maximizing the ELBO is equivalent to minimizing the KL divergence from $p(\boldsymbol{\theta}|\mathbf{x})$ to $q(\boldsymbol{\theta}; \phi)$. Writing the ELBO in terms of the transformed variables $\boldsymbol{\theta}^*$, we have:

$$\mathcal{L}(\phi, \zeta) = \mathbb{E}_{q_\phi(\boldsymbol{\theta}^*)}[\log p(\mathbf{x}, T^{-1}(\boldsymbol{\theta}^*); \zeta) + \log|\det J_{T^{-1}}(\boldsymbol{\theta}^*)| - \log q(\boldsymbol{\theta}^*; \phi)]. \tag{4}$$

The term $\log|\det J_{T^{-1}}(\boldsymbol{\theta}^*)|$ is a Jacobian adjustment for the transformation $T^{-1}$ that maps $\boldsymbol{\theta}^*$ to $\boldsymbol{\theta}$, required to ensure that the transformed density integrates to one. For the transformation $T^{-1}$, with $\boldsymbol{\theta}^*$ vectorized as $[b^*, A^*, t_0^*]$, the Jacobian is given by:

$$J_{T^{-1}}(\boldsymbol{\theta}^*) = \begin{bmatrix} b^* & A^* & 0 \\ 0 & A^* & 0 \\ 0 & 0 & t_0^* \end{bmatrix}.$$

Taking the log of the absolute value of the determinant of $J_{T^{-1}}(\boldsymbol{\theta}^*)$ gives the required adjustment:

$$\log|\det J_{T^{-1}}(\boldsymbol{\theta}^*)| = \log|b^* A^* t_0^*|. \tag{5}$$

We will apply stochastic gradient ascent to maximize the ELBO objective function given by *Equation 4*, using Monte Carlo (MC) estimates of the expectation and AD to calculate gradients with respect to $\phi$ and $\zeta$. However, the gradients of the ELBO with respect to $\phi$ cannot be calculated directly by AD, since the expectation in *Equation 4* is taken with respect to $q(\theta^*; \phi)$, which depends on $\phi$. We work around this by applying the *reparameterization trick* (*Kingma and Welling, 2013*; *Rezende et al., 2014*), which expresses $\theta^*$ as a differentiable transformation of an auxiliary noise variable $\epsilon \sim p(\epsilon)$, where $p(\epsilon)$ does not depend on $\phi$. In particular, we set $p(\epsilon)$ to be a standard multivariate Gaussian, $p(\epsilon) = \mathcal{N}(\epsilon; \mathbf{0}, \mathbf{I})$, and define the transformation $\widetilde{\theta}^* = \mathbf{L}\epsilon + \mu$, where μ and $\mathbf{L}$ are the mean and covariance parameters of the Gaussian approximating density $q(\theta^*; \phi)$.

Now we estimate the expectation in *Equation 4* with MC samples from $p(\epsilon)$. Since the expectation no longer depends on $\phi$, we can use AD directly on these MC estimates to obtain unbiased gradients of the ELBO. We estimate the ELBO for each data point using independent MC samples, since this reduces the variance of the gradients relative to using the same set of MC samples for a given batch of data (*Kingma et al., 2015*). Thus the ELBO objective for the *i*th data point is estimated by:

$$\widetilde{\mathcal{L}}^{(i)}(\phi, \zeta) = \frac{1}{L} \sum_{l=1}^{L} \log p(x^{(i)}, T^{-1}(\theta^{*(i,l)}); \zeta) + \log |\det J_{T^{-1}}(\theta^{*(i,l)})| - \log q(\theta^{*(i,l)}; \phi),$$

(6)

$$\text{where} \quad \theta^{*(i,l)} = \mathbf{L}\epsilon^{(i,l)} + \mu, \quad \epsilon^{(i,l)} \sim p(\epsilon), \quad \text{and} \quad x^{(i)} = \{t^{(i)}, c^{(i)}, s^{(i)}\}.$$

## VAM inference algorithm and other training details

The VAM training/inference algorithm is summarized in **Algorithm 1**. The size of the training set was 16,250 samples (65% of the total dataset) for each model. An additional validation set of 3750 samples (15%) was used to monitor model training. The remaining 5000 samples (20%) were used to evaluate model performance (the holdout set). We set the batch size $M$ to 256 and the number of MC samples used to estimate the ELBO $L$ to 10. We used the Adam optimizer *Kingma and Ba, 2017* with the following hyperparameters for model training: learning rate = 1e−3, $\beta_1 = 0.9$, and $\beta_2 = 0.999$. The same random seed was used to initialize the parameters of all models. Models were trained using a single NVIDIA GeForce RTX 3060 Ti GPU. Each model took approximately 1 hr to train.

For a small fraction of attempted model training runs (10/85=11.8%), the model either failed to exceed chance accuracy (3/10 models) or converged to a state in which almost all trials had exclusively negative drift rates (7/10 models). These models were not included in summary analyses.

---

**Algorithm 1. VAM inference algorithm.**

---

**Data:x**=\{RTs $t^{(i)} \ldots t^{(N)}$, choices $c^{(i)} \ldots c^{(N)}$, stimuli $s^{(i)} \ldots s^{(N)}$\}
$(\phi, \zeta) \leftarrow$ Initialize CNN parameters $\zeta$ and approximate posterior parameters $\phi$
**while** *not converged* **do**
  $\mathbf{t}^M, \mathbf{c}^M, \mathbf{s}^M \sim \mathbf{x}$ (sample random minibatch of size $M$ from full dataset)
  $\mathbf{v}^M \leftarrow \text{CNN}_\zeta(\mathbf{s}^M)$ (calculate drift rate means)
  $\epsilon^M \sim \mathcal{N}(\mathbf{0}, \mathbf{I})$ (draw $M \times L$ samples from the standard multivariate Gaussian)
  $\theta^{*M} \leftarrow \mathbf{L}\epsilon^M + \mu$ (reparameterize)
  $\widetilde{\mathcal{L}}^M(\phi, \zeta) \leftarrow$ calculate ELBO using *Equation 6*
  $\mathbf{g}^M \leftarrow \nabla_{\phi, \zeta} \widetilde{\mathcal{L}}^M(\phi, \zeta)$ (calculate gradients using AD)
  $(\phi, \zeta) \leftarrow$ update parameters using the Adam optimizer and $\mathbf{g}^M$
**end**

---

## Convolutional neural network (CNN)

We used the same seven-layer CNN architecture for all models (six convolutional layers followed by one fully-connected hidden layer). The number of channels/units used in the seven layers were as follows: 64, 64, 128, 128, 128, 256, 1024. The output layer (after the fully-connected layer) has four channels, one for each drift rate. Each convolutional layer used a 3×3 pixel kernel (stride 1, same padding). Each convolutional layer was followed by a ReLU nonlinearity, instance normalization (*Ulyanov et al., 2017*), and a max-pooling layer (2×2 pixel window, stride 2), in that order. The fully-connected hidden layer was followed by a ReLU nonlinearity and a dropout layer (dropout rate set to 0.5).

To speed up model training, the first two convolutional layers were initialized with the parameters from a larger CNN trained on an image classification task. Specifically, the pretrained model used a 16-layer VGG architecture and was trained on the ImageNet dataset (*Deng et al., 2009*; *Simonyan and Zisserman, 2015*). These first two layers were trainable (i.e. not fixed to their initial values). The weights of the other convolutional layers, fully-connected layer, and output layer were initialized randomly using the LeCun normal initializer *Klambauer et al., 2013*; the biases were initialized with zeros.

### Image preprocessing and data augmentation

The image stimuli used to train the VAM differed from the original Lost in Migration stimuli in a few ways. Information about the current score and time remaining visible at the top of the game window was removed. We also used the same blue background image for all stimuli (the background in the original Lost in Migration changes over the course of the gameplay). Finally, the stimuli were resized from 640×480 pixels (width×height) to 128×128 pixels.

We used data augmentation techniques commonly used in other image classification training paradigms to improve the generalization ability of the models. Specifically, for each batch of training data, each image was independently and randomly translated by a small amount. The size of the translation (in pixels) was drawn from a uniform distribution on the interval [0,1] (vertical) and [0,2] (horizontal), rounded to the nearest pixel (horizontal and vertical translations were sampled independently). We also applied a variation of a random elastic image deformation (*Simard et al., 2003*) as implemented in the Augmax Python package (*Heidler, 2022*). Specifically, we used the Warp transformation in Augmax and set the strength parameter to 3 and the coarseness parameter to 32. The transformation was applied to each image independently with a probability of 0.75.

### Task-optimized models

The task-optimized CNN models were trained by minimizing the cross-entropy loss function, where the correct label was defined as the true direction of the target bird in each stimulus (i.e. these models were not trained to match the decisions of the human participants, but rather to output the correct target direction). Other than the loss function, all aspects of the training process were the same as those used for the VAM (CNN architecture, optimizer settings, initialization, etc.). For each VAM we trained, we trained one task-optimized model using the same training data (*n*=75 task-optimized models).

### Analysis methods: Overview

We analyzed all of the VAMs after 12,800 parameter update steps (corresponding to the 200th training epoch), by which point training had converged. We analyzed all of the task-optimized models after 1600 parameter update steps (the 25th training epoch) since these models converged much more quickly. All analyses were conducted using a holdout set of $n$ = 5000 LIM stimuli/RTs/choices (i.e. data that was not used to train the models). To generate RTs and choices from the trained models, the LIM stimuli from the holdout set were provided as inputs to the CNN, which output mean drift rates for each stimulus. We denote the drift rate mean for the $k^{th}$ accumulator on trial $i$ as $v_k^{(i)}$. We used the LBA model to sample RTs and choices using these mean drift rates, where the drift rate for the $k^{th}$ accumulator on trial $i$, $d_k^{(i)}$, is sampled from a normal distribution with mean $v_k^{(i)}$ and SD = 1. For a very small fraction of trials (mean ± s.e.m. percent excluded trials: 0.080 ± 0.011%, $n$ = 75 models), all of the sampled drift rates were negative, and thus the RT and choice were undefined. These trials were excluded from all analyses.

All error bars shown in the figures correspond to bootstrap 95% confidence intervals calculated using 1000 bootstrap samples.

### Behavior analyses

The RT congruency effect was defined as the mean RT on incongruent trials minus the mean RT on congruent trials, where only correct trials were included in the calculation. The accuracy congruency effect was defined as the accuracy on congruent trials minus the accuracy on incongruent trials.

To calculate the RT delta plots for a given model/participant, we first calculated the RT deciles (0.1, 0.2, ..., 0.9 quantiles) separately for congruent and incongruent trials, using only correct trials. Within

each RT decile, we then calculated the RT congruency effect and mean RT (average of congruent and incongruent mean RTs), forming a delta plot for that participant/model. These mean RTs and congruency effects were then averaged across participants.

To calculate the conditional accuracy functions for a given model/participant, we calculated the accuracy and mean RT of trials within each RT quintile (0.2, 0.4, 0.6, and 0.8 quantiles) separately for congruent and incongruent trials. As above, these measures were then averaged across participants.

The models/participants included in the analysis of how RT varied with stimulus layout and horizontal/vertical position were those that exhibited significant modulation of RT by one or more of these stimulus features. Significant modulation was determined by running an ANOVA on the RTs in each stimulus feature bin (e.g. for each layout or each horizontal position bin), using a threshold $p$-value of 0.05. For horizontal stimulus position, we used bins of width = 50 pixels in the original 640×480 pixel window space, except for the leftmost and rightmost bins which had width = 60 pixels. For vertical stimulus position, we used bins of width = 25 pixels.

The number of participants exhibiting significant modulation of RT was 60 for stimulus layout, 72 for horizontal position, and 69 for vertical position (out of 75 total). To determine whether a given participant exhibited significant modulation of accuracy by a given stimulus feature, we used a chi-squared test for equality of proportions across stimulus feature bins (threshold p-value=0.05). No participants exhibited significant modulation of accuracy for any of the stimulus features.

## LBA parameters

For analyses of the fitted LBA parameters $t_0$, $b$, and $A$, we used the maximum a posteriori (MAP) estimates of these parameters, corresponding to the mean parameter vector of the learned Gaussian posterior density $q(\boldsymbol{\theta}; \phi)$. For analyses of the mean target and flanker drift rates, we provided the LIM stimuli from the holdout set as inputs to the fitted CNN, which generates the mean drift rates as its outputs. The non-target/non-flanker (other) drift rates were calculated by averaging the drift rates from the two non-target/non-flanker accumulators on incongruent trials or the three non-target accumulators on congruent trials.

## CNN unit activity

The activation matrices used in the analyses of CNN representations were derived from the responses of units in each layer elicited by the holdout image set. The activations were processed just after the ReLU nonlinearity. For the convolutional layers, we defined the activation of a given unit/channel as the maximum value of that channel calculated across the spatial dimensions (*Simonyan and Zisserman, 2015*). This yields a $N \times K_l$ activation matrix, where $N$ is the number of images in the holdout set (5000) and $K_l$ is the number of active channels in layer $l$. For the fully-connected layer, the responses of all units were used to define the $N \times K_l$ activation matrix directly.

Only incongruent trials were used for all of the analyses involving the CNN activations described below, for the following reasons. For the selectivity and tolerance analyses in which we decoded target or flanker direction from the activity in each layer, described in more detail below, note that the target and flanker directions are by definition the same on congruent trials. As such, a classifier trained to decode target direction from congruent trials could achieve perfect accuracy using information in the unit activity attributable to the flankers, rather than targets. Using only incongruent trials ensured that such cross-contamination could not occur. We used only incongruent trials for all of the other analyses of the activations simply for convenience.

## Stimulus feature decoding

To assess how well particular stimulus features could be decoded from the activity in each layer, we trained a linear SVM to classify the values of that stimulus feature using the $N \times K_l$ activation matrix in each layer. The activation matrix was standardized before training the classifiers, such that each column had zero mean and unit variance. Horizontal and vertical stimulus positions were discretized to enable classification using the same bins as were used in the behavioral analyses.

The classification task was done in a standard 'one-vs-rest' setting: for each value of a given stimulus feature, one sub-classifier was trained on a binary classification task with the chosen value as one class and all other values as the other class, yielding one classifier for each value of the stimulus feature (e.g. four for target direction, seven for stimulus layout). To determine the decision of the

combined classifier for a given image, we generated predictions from each sub-classifier and assigned the decision to the sub-classifier with the largest (most confident) prediction. We assessed the overall decoding accuracy of each SVM on a separate test image set. Note that both the training set and test set for the SVMs were subsets of the holdout image set (i.e. the images that were not used to train the VAMs). The SVMs were trained using the LinearSVC model in the scikit-learn Python package *Pedregosa, 2011* with the squared hinge loss function and L2 regularization with the penalty parameter C set to 1.0 (the default settings).

### Tolerance

To assess the tolerance of model representations to variation in a given stimulus feature (flanker direction, stimulus layout, horizontal position, and vertical position), we trained a linear SVM to classify target direction from the CNN activations using stimuli with the chosen stimulus feature fixed to one value (the training context), and assessed the generalization performance of the SVM on stimuli that contained all other values of that stimulus feature (the generalization context). For example, to assess tolerance to stimulus layout, we trained one SVM to classify target direction using stimuli with the vertical line layout, and assessed the generalization performance of that classifier on stimuli with the six other layouts. We trained one such SVM for each of the seven stimulus layouts, and averaged the generalization performance across these seven SVMs to derive the overall generalization performance measure for a given model and network layer. Other details of the classifiers were the same as those used for the decoding analyses described above.

### Target/flanker subspace alignment

To calculate the target/flanker subspace alignment metric, we first defined target and flanker subspaces using the SVM classifiers that we trained for the target/flanker decoding analyses. Specifically, for each of the four classifiers, which were trained to classify stimuli as a given target or flanker direction vs. all other target or flanker directions using the CNN activations from a given layer, we extracted the vector orthogonal to the decision hyperplane. In agreement with prior work (*Bernardi et al., 2020*; *Libby and Buschman, 2021*), we refer to these vectors as decoding vectors for a given target/flanker direction. Let $\mathbf{x}_{k,\mathrm{targ}}^T$ and $\mathbf{x}_{k,\mathrm{flnk}}^T$ denote the target decoding row vector and flanker decoding row vector for the $k^{\mathrm{th}}$ direction, respectively. We define the matrices formed with these four decoding vectors filling the rows as the target subspace matrix $\mathbf{X}_{\mathrm{targ}}$ and the flanker subspace matrix $\mathbf{X}_{\mathrm{flnk}}$. These matrices have dimensions $4 \times K_l$, where $K_l$ is the number of active units in layer $l$. Each matrix, therefore, spans a subspace of the full $K_l$-dimensional space. Our goal is to determine whether the target and flanker subspaces are orthogonal.

To do so, we employ principal angles between subspaces (*Jordan, 1873*; *Zhu and Knyazev, 2013*), which generalizes the more intuitive notion of angles between lines or planes to arbitrary dimensions. To calculate the principal angles, we require orthonormal bases for the target and flanker subspaces, which we determine using a reduced singular value decomposition (SVD):

$$\mathbf{X}_{\mathrm{targ}} = \mathbf{U}_{\mathrm{targ}} \boldsymbol{\Sigma}_{\mathrm{targ}} \mathbf{V}_{\mathrm{targ}}^T, \qquad \mathbf{X}_{\mathrm{flnk}} = \mathbf{U}_{\mathrm{flnk}} \boldsymbol{\Sigma}_{\mathrm{flnk}} \mathbf{V}_{\mathrm{flnk}}^T.$$

The rows of the $4 \times K_l$ matrices $\mathbf{V}_{\mathrm{targ}}^T$ and $\mathbf{V}_{\mathrm{flnk}}^T$ form an orthonormal basis for the target and flanker subspaces, respectively. The cosines of the principal angles are given by the singular values of $\mathbf{V}_{\mathrm{targ}} \mathbf{V}_{\mathrm{flnk}}^T$. The average of these singular values is our subspace alignment metric, which ranges from zero (completely orthogonal subspaces) to one (completely aligned/parallel subspaces).

### Participation ratio

We measured the dimensionality of target representations for a given layer with the participation ratio ($\mathrm{PR}_l$) (*Gao et al., 2017*), defined as:

$$\mathrm{PR}_l = \frac{\left( \sum_{i=1}^{K_l} \lambda_i \right)^2}{\sum_{i=1}^{K_l} \lambda_i^2},$$

where $K_l$ is the number of active units in layer $l$ and $\lambda_1 \geq \ldots \geq \lambda_i \geq \ldots \geq \lambda_{K_l}$ are the eigenvalues of the target-centered activation covariance matrix for layer $l$. The target-centered activations were

obtained by subtracting the centroid of the activation matrix for each target direction from the corresponding trials in the activation matrix (***Rangamani et al., 2023***).

The participation ratio is a continuous measure of dimensionality ranging from 1 to $K_l$. The minimum ($\mathrm{PR}_l = 1$) is obtained when all of the variance in activity is concentrated in a single dimension, such that $\lambda_i = 0$ for $i \geq 2$. The maximum ($\mathrm{PR}_l = K_l$) is obtained when the variance is evenly spread across the $K_l$ dimensions, such that all $K_l$ eigenvalues are equal.

## Mutual information

The activity of each unit in response to the holdout image set was discretized into 10 equally-sized bins, yielding a unit-specific activation distribution $p_X(x)$. Stimulus features (horizontal/vertical position, layout, target/flanker direction) were discretized if the values were continuous, or used as is if not, yielding a stimulus feature distribution $p_Y(y)$. Discretization of the continuous variables was done as described in the decoding methods section. The joint probability mass function of the unit activity and stimulus feature is denoted by $p_{(X,Y)}(x, y)$. For a given unit and stimulus feature, the mutual information is given by:

$$I(X; Y) = \sum_{x,y} p_{(X,Y)}(x, y) \log \frac{p_{(X,Y)}(x, y)}{p_X(x) p_Y(y)}.$$

The mutual information for a given stimulus feature was normalized by the entropy of the feature distribution to facilitate comparisons between the features ***Muratore et al., 2022***. The entropy is given by:

$$H(Y) = -\sum_y p_Y(y) \log p_Y(y).$$

## Single-unit modulation by target direction

To identify units that were modulated by target direction, we ran a one-way ANOVA on the z-scored activity of each unit, where each group in the ANOVA was determined by the activity of that unit in response to stimuli for a given target direction (the activity of each unit was z-scored across the stimuli). Units with an ANOVA $p$-value<0.001 were defined as significantly modulated by target direction. These units were further split into three subtypes based on their degree of selectivity and sign of modulation: selective (+), selective (-), and complex units.

We first selected units that had significantly higher or lower activation for one direction relative to the other three directions, as assessed by Tukey's HSD test ($p$<0.05). Within this population, some units had both significantly higher activation for one direction relative to the other three *and* significantly lower activation for one direction relative to the other three. Units that did vs. did not have this property were handled separately. In the former (simpler) case, the units that only had significantly higher activation for one direction relative to the other three were defined as selective (+) units; the units that only had significantly lower activation for one direction relative to the other three were defined as selective (-) units.

In the latter (more complicated) case, we compared the magnitude of the activation for the positive and negative modulation directions with a rank-sum test. Units for which the magnitude of activity for the positive modulation direction was significantly greater ($p$<0.05) than the magnitude of activity for the negative modulation direction were defined as selective (+) units. Analogous criteria were used to define selective (-) units (with the signs reversed). Units within this pool with rank-sum p-value>0.05 were defined as complex units. Finally, units that were significantly modulated by target direction but that did not meet any of the criteria described above were also defined as complex units.

## Acknowledgements

We thank M Steyvers, M Robinson, D Yamins and members of the NeuroAILab, the Lumos Labs research team, and members of the Poldrack Lab for helpful conversations and comments on this work. We also thank A Kaluszka for providing graphics files of the Lost and Migration stimuli.

## Additional information

### Funding
No external funding was received for this work.

### Author contributions
Paul I Jaffe, Conceptualization, Data curation, Software, Formal analysis, Validation, Investigation, Visualization, Methodology, Writing – original draft, Writing – review and editing; Gustavo X Santiago-Reyes, Software, Formal analysis, Investigation, Methodology, Writing – review and editing; Robert J Schafer, Russell A Poldrack, Conceptualization, Resources, Supervision, Methodology, Project administration, Writing – review and editing; Patrick G Bissett, Conceptualization, Supervision, Methodology, Project administration, Writing – review and editing

### Author ORCIDs
Paul I Jaffe ⬤ https://orcid.org/0000-0003-0680-3923
Patrick G Bissett ⬤ https://orcid.org/0000-0003-0854-9404

Reviewer #1 (Public review): https://doi.org/10.7554/eLife.98351.3.sa1
Reviewer #2 (Public review): https://doi.org/10.7554/eLife.98351.3.sa2
Author response https://doi.org/10.7554/eLife.98351.3.sa3

---

## Additional files

### Supplementary files
MDAR checklist

### Data availability
All of the code (https://github.com/pauljaffe/vam, copy archived at *Jaffe, 2025*) and data (https://doi.org/10.5281/zenodo.10775513) used to train the VAMs and reproduce our results are publicly-available without restrictions.

The following dataset was generated:

| Author(s) | Year | Dataset title | Dataset URL | Database and Identifier |
|---|---|---|---|---|
| Jaffe PI, Gustavo XSR, Schafer RJ, Bissett PG, Poldrack RA | 2025 | Data and models for 'An image-computable model of speeded decision-making' | https://doi.org/10.5281/zenodo.10775513 | Zenodo, 10.5281/zenodo.10775513 |

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
