## [Editor Report · eLife Assessment]

This **important** study presents an original and promising approach to combine convolutional neural networks of visual processing with evidence accumulation models of decision-making. While the methodological approach is technically sophisticated and the evidence is **solid**, there is still a gap between the model and the behavioral data. The study will be of interest to researchers working in the fields of machine learning and cognitive modeling.

---

## [Referee Report · Reviewer #1 (Public review)]

Summary:

This paper introduces a new approach for modeling human behavioral responses using image-computable models. They create a model (VAM) that is a combination of a standard CNN coupled with a standard evidence accumulation model (EAM). The combined model is then trained directly on image-level data using human behavioral responses. This approach is original and can have wide applicability. However, many of the specific findings reported are less compelling.

Strengths:

(1) The manuscript presents an original approach of fitting an image-computable model to human behavioral data. This type of approach is sorely needed in the field.

(2) The analyses are very technically sophisticated.

(3) The behavioral data are large both in terms of sample size (N=75) and in terms of trials per subject.

Weaknesses:

(1) The main advance here thus appears to be methodological rather than conceptual. It's really cool that VAMs are image computable and are also fit to human data. But what we learn about the mind or brain is perhaps more modest.

(2) In the approach here, a given stimulus is always processed in the same way through the core CNN to produce activations v_k. These v_k's are then corrupted by Gaussian noise to produce drift rates d_k, which can differ from trial to trial even for the same stimulus. In other words, the assumption built into VAM appears to be that the drift rate variability stems entirely from post-sensory (decisional) noise. In contrast, the typical interpretation of EAMs is that the variability in drift rates is sensory. In response to this concern, the authors responded that one can imagine an additional (unmodeled) sensory process that adds variability to the drift rates. However, this process remains unmodeled. The authors motivate their paper by saying "EAMs do not explain how the visual system extracts these representations in the first place" (second sentence of the Abstract). VAM is definitely a step in this direction but there's still a gap between the current VAM implementation and sensory systems.

---

## [Referee Report · Reviewer #2 (Public review)]

In An image-computable model of speeded decision-making, the authors introduce and fit a combined CCN-EAM (a 'VAM') to flanker-task-like data. They show that the VAM can fit mean RTs and accuracies as well as the congruency effect that is present in the data, and subsequently analyze the VAM in terms of where in the network congruency effects arise.

I have mixed feelings about this manuscript, as I appreciate the innovative efforts to combine CNNs with EAMs in a new class of cognitive models, while also having some reservations from an EAM perspective. The idea of combining these approaches has great potential, and I'm excited to see where this research will lead. However, I do have some concerns about the quality of fit between the behavioral data and the model. Specifically, the RT distributions, delta plots, and conditional accuracy function don't appear to be well-matched by the VAM. The conflict effects on behavioral data are well-established and typically considered crucial to understanding the underlying cognitive process. Unfortunately, it seems that these parts of the data don't fit well with the proposed model.

This disparity is not entirely surprising. The EAM literature suggests that LBA models might not be suitable for conflict tasks, and the presented results seem to confirm this concern. Conflict EAMs, including the DMC (e.g., Ulrich et al., 2015; Evans & Servant, 2022; Lee & Sewell 2024), propose dynamic drift rates with a fast automatic process that is gradually withdrawn from evidence accumulation over time. This approach results in congruency effects arising from temporal dynamics, not spatial representations.

In contrast, the VAM imposes static drift rates in the LBA model, leading to an effect between drift rates that translates to changes in representations. However, this account does not adequately explain the behavioral data, and the proposed representational geometry explanation is therefore limited.

My concerns are addressed in the revised manuscript, but I struggle to understand why the authors distinguish between explaining mean effects across individuals and congruency effects within individuals. These concepts seem related, and issues at the individual level could propagate to the group mean. Furthermore, I find it challenging to accept that dynamics merely act 'in concert' with the orthogonalization mechanism, as it seems possible that an account that uses a time-varying EAM may not require any orthogonalization mechanism in the first place. The orthogonalization mechanism might have arisen because the model does not have the possibility to account for the conflict effect from temporal effects, instead of spatial effects. I could envision a CNN-DMC in which conflict effects arise only at the level of the choice model (e.g., as a time-varying filter that changes which information is read out from the visual system, rather than due to changes in the representations in the visual system itself). This possibility should be acknowledged in the paper, and it would be interesting to discuss how such an account would be tested.

While I appreciate the technological advancement presented in this paper, my concerns are not about implementation details but rather about the choice of models and their consequences. I believe that a more in-depth exploration of which conclusions can be drawn, and which model comparisons would be required to reach a final conclusion.

---

## [Author Response]

The following is the authors’ response to the original reviews.

**Public Reviews:**

**Reviewer #1 (Public Review):**
Summary:This paper introduces a new approach to modeling human behavioral responses using image-computable models. They create a model (VAM) that is a combination of a standard CNN coupled with a standard evidence accumulation model (EAM). The combined model is then trained directly on image-level data using human behavioral responses. This approach is original and can have wide applicability. However, many of the specific findings reported are less compelling.Strengths:(1) The manuscript presents an original approach to fitting an image-computable model to human behavioral data. This type of approach is sorely needed in the field.(2) The analyses are very technically sophisticated.(3) The behavioral data are large both in terms of sample size (N=75) and in terms of trials per subject.Weaknesses:Major(1) The manuscript appears to suggest that it is the first to combine CNNs with evidence accumulation models (EAMs). However, this was done in a 2022 preprint(https://www.biorxiv.org/content/10.1101/2022.08.23.505015v1) that introduced a network called RTNet. This preprint is cited here, but never really discussed. Further, the two unique features of the current approach discussed in lines 55-60 are both present to some extent in RTNet. Given the strong conceptual similarity in approach, it seems that a detailed discussion of similarities and differences (of which there are many) should feature in the Introduction.

Thanks for pointing this out—we agree that the novel contributions of our model (the VAM) with respect to prior related models (including RTNet) should be clarified, and have revised the Introduction accordingly. We include the following clarifications in the Introduction:

“The key feature of the VAM that distinguishes it from prior models is that the CNN and EAM parameters are jointly fitted to the RT, choice, and visual stimulus data from individual participants in a unified Bayesian framework. Thus, both the visual representations learned by the CNN and the EAM parameters are directly constrained by behavioral data. In contrast, prior models first optimize the CNN to perform the behavioral task, then separately fit a minimal set of high-level CNN parameters [RTNet, Rafiei et al., 2024] and/or the EAM parameters to behavioral data [Annis et al., 2021; Holmes et al., 2020; Trueblood et al., 2021]. As we will show, fitting the CNN with human data—rather than optimizing the model to perform a task—has significant consequences for the representations learned by the model.”

E.g. in the case of RTNet, the variability of the Bayesian CNN weight distribution, the decision threshold, and the magnitude of the noise added to the images are adjusted to match the average human accuracy (separately for each task condition). RTNet is an interesting and useful model that we believe has complementary strengths to our own work.

Since there are several other existing models in addition to the VAM and RTNet that use CNNs to generate RTs or RT proxies (by our count, at least six that we cite earlier in the Introduction), we felt it was inappropriate to preferentially include a detailed comparison of the VAM and RTNet beyond the passage quoted above.

(2) In the approach here, a given stimulus is always processed in the same way through the core CNN to produce activations v_k. These v_k's are then corrupted by Gaussian noise to produce drift rates d_k, which can differ from trial to trial even for the same stimulus. In other words, the assumption built into VAM appears to be that the drift rate variability stems entirely from post-sensory (decisional) noise. In contrast, the typical interpretation of EAMs is that the variability in drift rates is sensory. This is also the assumption built into RTNet where the core CNN produces noisy evidence. Can the authors comment on the plausibility of VAM's assumption that the noise is post-sensory?

In our view, the VAM is compatible with a model in which the drift rate variability for a given stimulus is due to sensory noise, since we do not specify the origin of the Gaussian noise added to the drift rates. As the reviewer notes, the CNN component of the VAM processes a given stimulus deterministically, yielding the mean drift rates. This does not preclude us from imagining an additional (unmodeled) sensory process that adds variability to the drift rates. The VAM simply represents this and other hypothetical sources of variability as additive Gaussian noise. We agree however that it is worthwhile to think about the origin of the drift rate variability, though it is not a focus of our work.

(3) Figure 2 plots how well VAM explains different behavioral features. It would be very useful if the authors could also fit simple EAMs to the data to clarify which of these features are explainable by EAMs only and which are not.

In our view, fitting simple EAMs to the data would not be especially informative and poses a number of challenges for the particular task we study (LIM) that are neatly avoided by using the VAM. In particular, as we show in Figure 2, the stimuli vary along several dimensions that all appear to influence behavior: horizontal position, vertical position, layout, target direction, and flanker direction. Since the VAM is stimulus-computable, fitting the VAM automatically discovers how all of these stimulus features influence behavior (via their effect on the drift rates outputted by the CNN). In contrast, fitting a simple EAM (e.g. the LBA model) necessitates choosing a particular parameterization that specifies the relationship between all of the stimulus features and the EAM model parameters. This raises a number of practical questions. For example, should we attempt to fit a separate EAM for each stimulus feature, or model all stimulus features simultaneously?

Moreover, while we could in principle navigate these issues and fit simple EAMs to the data, we do not intend to claim that simple EAMs fail to explain the relationship between stimulus features and behavior as well as the VAM. Rather, the key strength of the VAM relative to simple EAMs is that it includes a detailed and biologically plausible model of human vision. The majority of the paper capitalizes on this strength by showing how behavioral effects of interest (namely congruency effects) can be explained in terms of the VAM’s visual representations.

(4) VAM is tested in two different ways behaviorally. First, it is tested to what extent it captures individual differences (Figure 2B-E). Second, it is tested to what extent it captures average subject data (Figure 2F-J). It wasn't clear to me why for some metrics only individual differences are examined and for other metrics only average human data is examined. I think that it will be much more informative if separate figures examine average human data and individual difference data. I think that it's especially important to clarify whether VAM can capture individual differences for the quantities plotted in Figures 2F-J.

We would like to clarify that Fig. 2J in fact already shows how well the VAM captures individual differences for the average subject data shown in Fig. 2H (stimulus layout) and Fig. 2I (stimulus position). For a given participant and stimulus feature, we calculated the Pearson's r between model/participant mean RTs across each stimulus feature value. Fig. 2J shows the distribution of these Pearson’s r values across all participants for stimulus layout and horizontal/vertical position.

Fig. 2G also already shows how well the VAM captures individual differences in behavior. Specifically, this panel shows individual differences in mean RT attributable to differences in age. For Fig. 2F, which shows how the model drift rates differ on congruent vs. incongruent trials, there is no sensible way to compare the models to the participants at any level of analysis (since the participants do not have drift rates).

(5) The authors look inside VAM and perform many exploratory analyses. I found many of these difficult to follow since there was little guidance about why each analysis was conducted. This also made it difficult to assess the likelihood that any given result is robust and replicable. More importantly, it was unclear which results are hypothesized to depend on the VAM architecture and training, and which results would be expected in performance-optimized CNNs. The authors train and examine performance-optimized CNNs later, but it would be useful to compare those results to the VAM results immediately when each VAM result is first introduced.

Thanks for pointing this out—we apologize for any confusion caused by our presentation of the CNN analyses. We have added in additional motivating statements, methodological clarifications, and relevant references to our Results, particularly for Figure 3 in which we first introduce the analyses of the CNN representations/activity. In general, each analysis is prefaced by a guiding question or specific rationale, e.g. “How do the models' visual representations enable target selectivity for stimuli that vary along several irrelevant dimensions?” We also provide numerous references in which these analysis techniques have been used to address similar questions in CNNs or the primate visual cortex.

We chose to maintain the current organization of our results in which the comparison between the VAM and the task-optimized models are presented in a separate figure. We felt that including analyses of both the VAM and task-optimized models in the initial analyses of the CNN representations would be overwhelming for many readers. As the reviewer acknowledges, some readers may already find these results challenging to follow.

(6) The authors don't examine how the task-optimized models would produce RTs. They say in lines 371-2 that they "could not examine the RT congruency effect since the task-optimized models do not generate RTs." CNNs alone don't generate RTs, but RTs can easily be generated from them using the same EAM add-on that is part of VAM. Given that the CNNs are already trained, I can't see a reason why the authors can't train EAMs on top of the already trained CNNs and generate RTs, so these can provide a better comparison to VAM.

We appreciate this suggestion, but we judge the suggestion to “train EAMs on top of the already trained CNNs and generate RTs” to be a significant expansion of the scope of the paper with multiple possible roads forward. In particular, one must specify how the outputs of the task-optimized CNN (logits for each possible response) relate to drift rates, and there is no widely-accepted or standard way to do this. Previously proposed methods include transforming representation distances in the last layer to drift rates (https://doi.org/10.1037/xlm0000968), fitting additional subject-specific parameters that map the logits to drift rates

(https://doi.org/10.1007/s42113-019-00042-1), or using the softmax-scored model outputs as drift rates directly (https://doi.org/10.1038/s41562-024-01914-8), though in the latter case the RTs are not on the same scale as human data. In our view, evaluating these different methods is beyond the scope of this paper. An advantage of the VAM is that one does not have to fit two separate models (a CNN and a EAM) to generate RTs.

Nonetheless, we agree that it would be informative to examine something like RTs in the task-optimized models. Our revised Results section now includes an analysis of the confidence of the task-optimized models’ decisions, which we use a proxy for RTs:

“Since the task-optimized models do not generate RTs, it is not possible to directly measure RT congruency effects in these models without making additional assumptions about how the CNN's classification decisions relate to RTs. However, as a coarse proxy for RT, we can examine the confidence of the CNN's decisions, defined as the softmax-scored logit (probability) of the most probable direction in the final CNN layer. This choice of RT proxy is motivated by some prior studies that have combined CNNs with EAMs [Annis et al., 2021; Holmes et al., 2020; Trueblood et al., 2021]. These studies explicitly or implicitly derive a measure of decision confidence from the activity of the last CNN layer. The confidence measure is then mapped to the EAM drift rates, such that greater decision confidence generally corresponds to higher drift rates (and therefore shorter RTs).

We calculated the average confidence of each task-optimized CNN separately for congruent vs. incongruent trials. On average, the task-optimized models showed higher confidence on congruent vs. incongruent trials (W = 21.0, p < 1e-3, Wilcoxon signed-rank test; Cohen's d = 0.99; n = 75 models). These analyses therefore provide some evidence that task-optimized CNNs have the capacity to exhibit congruency effects, though an explicit comparison of the magnitude of these effects with human data requires additional modeling assumptions (e.g., fitting a separate EAM).”

(7) The Discussion felt very long and mostly a summary of the Results. I also couldn't shake the feeling that it had many just-so stories related to the variety of findings reported. I think that the section should be condensed and the authors should be clearer about which explanations are speculations and which are air-tight arguments based on the data.

We have shortened the Discussion modestly and we have added in some clarifying language to help clarify which arguments are more speculative vs. directly supported by our data.

Specifically, we added in the phrase “we speculate that…” for two suggestions in the Discussion (paragraphs 3 and 5), and we ensured that any other more speculative suggestions contain such clarifying language. We have also added in subheadings in the Discussion to help readers navigate this section.

(8) In one of the control analyses, the authors train different VAMs on each RT quantile. I don't understand how it can be claimed that this approach can serve as a model of an individual's sensory processing. Which of the 5 sets of weights (5 VAMs) captures a given subject's visual processing? Are the authors saying that the visual system of a given subject changes based on the expected RT for a stimulus? I feel like I'm missing something about how the authors think about these results.

We agree that these particular analyses may cause confusion and have removed them from our revised manuscript.

**Reviewer #2 (Public Review):**
In an image-computable model of speeded decision-making, the authors introduce and fit a combined CCN-EAM (a 'VAM') to flanker-task-like data. They show that the VAM can fit mean RTs and accuracies as well as the congruency effect that is present in the data, and subsequently analyze the VAM in terms of where in the network congruency effects arise.Overall, combining DNNs and EAMs appears to be a promising avenue to seriously model the visual system in decision-making tasks compared to the current practice in EAMs. Some variants have been proposed or used before (e.g., doi.org/10.1016/j.neuroimage.2017.12.078 , doi.org/10.1007/s42113-019-00042-1), but always in the context of using task-trained models, rather than models trained on behavioral data. However, I was surprised to read that the authors developed their model in the context of a conflict task, rather than a simpler perceptual decision-making task. Conflict effects in human behavior are particularly complex, and thereby, the authors set a high goal for themselves in terms of the to-be-explained human behavior. Unfortunately, the proposed VAM does not appear to provide a great account of conflict effects that are considered fundamental features of human behavior, like the shape of response time distributions, and specifically, delta plots (doi.org/10.1037/0096-1523.20.4.731). The authors argue that it is beyond the scope of the presented paper to analyze delta plots, but as these are central to studies of human conflict behavior, models that aim to explain conflict behavior will need to be able to fit and explain delta plots.Theories on conflict often suggest that negative/positive-trending delta plots arise through the relative timing of response activation related to relevant and irrelevant information.Accumulation for relevant and irrelevant information would, as a result, either start at different points in time or the rates vary over time. The current VAM, as a feedforward neural network model, does not appear to be able to capture such effects, and perhaps fundamentally not so: accumulation for each choice option is forced to start at the same time, and rates are a static output of the CNN.The proposed solution of fitting five separate VAMs (one for each of five RT quantiles) is not satisfactory: it does not explain how delta plots result from the model, for the same reason that fitting five evidence accumulation models (one per RT quantile) does not explain how response time distributions arise. If, for example, one would want to make a prediction about someone's response time and choice based on a given stimulus, one would first have to decide which of the five VAMs to use, which is circular. But more importantly, this way of fitting multiple models does not explain the latent mechanism that underlies the shape of the delta plots.As such, the extensive analyses on the VAM layers and the resulting conclusions that conflict effects arise due to changing representations across layers (e.g., "the selection of task-relevant information occurs through the orthogonalization of relevant and irrelevant representations") - while inspiring, they remain hard to weigh, as they are contingent on the assumption that the VAM can capture human behavior in the conflict task, which it struggles with. That said, the promise of combining CNNs and EAMs is clearly there. A way forward could be to either adjust the proposed model so that it can explain delta plots, which would potentially require temporal dynamics and time-varying evidence accumulation rates, or perhaps to start simpler and combine CCNs-EAMs that are able to fit more standard perceptual decision-making tasks without conflict effects.

We thank the reviewer for their thoughtful comments on our work. However, we note that the

VAM does in fact capture the positive-trending RT delta plot observed in the participant data (Fig. S4A), though the intercepts for models/participants differ somewhat. On the other hand, the conditional accuracy functions (Fig. S4B) reveal a more pronounced difference between model and participant behavior. As the reviewer points out, capturing these effects is likely to require a model that can produce time-varying drift rates, whereas our model produces a fixed drift rate for a given stimulus. We also agree that fitting a separate VAM to each RT quantile is not a satisfactory means of addressing this limitation and have removed these analyses from our revised manuscript.

However, while we agree that accurately capturing these dynamic effects is a laudable goal, it is in our view also worthwhile to consider explanations for the mean behavioral effect (i.e. the accuracy congruency effect), which can occur independently of any consideration of dynamics. One of our main findings is that across-model variability in accuracy congruency effects is better attributed to variation in representation geometry (target/flanker subspace alignment) vs.

variation in the degree of flanker suppression. This finding does not require any consideration of dynamics to be valid at the level of explanation we pursue (across-user variability in congruency effects), but also does not preclude additional dynamic processes that could give rise to more specific error patterns. Our revised discussion now includes a section where we summarize and elaborate on these ideas:

“It is not difficult to imagine how the orthogonalization mechanism described above, which explains variability in accuracy congruency effects across individuals, could act in concert with other dynamic processes that explain variability in congruency effects within individuals (e.g., as a function of RT). In general, any process that dynamically gates the influence of irrelevant sensory information on behavioral outputs could accomplish this, for example ramping inhibition of incorrect response activation [https://doi.org/10.3389/fnhum.2010.00222], a shrinking attention spotlight [https://doi.org/10.1016/j.cogpsych.2011.08.001], or dynamics in neural population-level geometry [https://doi.org/10.1038/nn.3643]. To pursue these ideas, future work may aim to incorporate dynamics into the visual component and decision component of the VAM with recurrent CNNs [https://doi.org/10.48550/arXiv.1807.00053, https://doi.org/10.48550/arXiv.2306.11582] and the task-DyVA model [https://doi.org/10.1038/s41562-022-01510-8], respectively.”

**Reviewer #3 (Public Review):**
Summary:In this article, the authors combine a well-established choice-response time (RT) model (the Linear Ballistic Accumulator) with a CNN model of visual processing to model image-based decisions (referred to as the Visual Accumulator Model - VAM). While this is not the first effort to combine these modeling frameworks, it uses this combination of approaches uniquely.Specifically, the authors attempt to better understand the structure of human information representations by fitting this model to behavioral (choice-RT) data from a classic flanker task. This objective is made possible by using a very large (by psychological modeling standards) industry data set to jointly fit both components of this VAM model to individual-level data. Using this approach, they illustrate (among other results) (1) how the interaction between target and flanker representations influence the presence and strength of congruency effects, (2) how the structure of representations changes (distributed versus more localized) with depth in the CNN model component, and (3) how different model training paradigms change the nature of information representations. This work contributes to the ML literature by demonstrating the value of training models with richer behavioral data. It also contributes to cognitive science by demonstrating how ML approaches can be integrated into cognitive modeling. Finally, it contributes to the literature on conflict modeling by illustrating how information representations may lead to some of the classic effects observed in this area of research.Strengths:(1) The data set used for this analysis is unique and is made publicly available as part of this article. Specifically, they have access to data for 75 participants with >25,000 trials per participant. This scale of data/individual is unusual and is the foundation on which this research rests.(2) This is the first time, to my knowledge, that a model combining a CNN with a choice-RT model has been jointly fit to choice-RT data at the level of individual people. This type of model combination has been used before but in a more restricted context. This joint fitting, and in particular, learning a CNN through the choice-RT modeling framework, allows the authors to probe the structure of human information representations learned directly from behavioral data.(3) The analysis approaches used in this article are state-of-the-art. The training of these models is straightforward given the data available. The interesting part of this article (opinion of course) is the way in which they probe what CNN has learned once trained. I find their analysis of how distractor and target information interfere with each other particularly compelling as well as their demonstration that training on behavioral data changes the structure of information representations when compared to training models on standard task-optimized data.Weaknesses:(1) Just as the data in this article is a major strength, it is also a weakness. This type of modeling would be difficult, if not impossible to do with standard laboratory data. I don't know what the data floor would be, but collecting tens of thousands of decisions for a single person is impractical in most contexts. Thus this type of work may live in the realm of industry. I do want to re-iterate that the data for this study was made publicly available though!

We suspect (but have not systematically tested) that the VAMs can be fitted with substantially less data. We use data augmentation techniques (various randomized image transformations) during training to improve the generalization capabilities of the VAMs, and these methods are likely to be particularly important when training on smaller datasets. One could consider increasing the amount of image data augmentation when working with smaller datasets, or pursuing other forms of data augmentation like resampling from estimated RT distributions (see https://doi.org/10.1038/s41562-022-01510-8 for an example of this). In general, we don’t think that prospective users of our approach should be discouraged if they have only a few hundred trials per subject (or less) - it’s worth trying!

(2) While this article uses choice-RT data it doesn't fully leverage the richness of the RT data itself. As the authors point out, this modeling framework, the LBA component in particular, does not account for some of the more nuanced but well-established RT effects in this data. This is not a big concern given the already nice contributions of this article and it leads to an opportunity for ongoing investigation.

We agree that fully capturing the more nuanced behavioral effects you mention (e.g. RT delta plots and conditional accuracy functions) is a worthwhile goal for future research—see our response to Reviewer #2 for a more detailed discussion. ----------

**Recommendations for the authors:**

**Reviewer #1 (Recommendations For The Authors):**
(1) The phrase in the Abstract "convolutional neural network models of visual processing and traditional EAMs are jointly fitted" made me initially believe that the two models were fitted independently. You may want to re-word to clarify.

We think that the phrase “jointly fitted” already makes it clear that both the CNN and EAM parameters are estimated simultaneously, in agreement with how this term is usually used. But we have nonetheless appended some additional clarifying language to that sentence (“in a unified Bayesian framework”).

(2) Lines 27-28: EAMs "are the most successful and widely-used computational models of decision-making." This is only true for the specific type of decision-making examined here, namely joint modeling of choice and response times. Signal detection theory is arguably more widely-used when response times are not modeled.

Thanks for pointing this out - we have revised the referenced sentence accordingly.

(3) Could the authors clarify what is plotted in Figure 2F?

Fig. 2F shows the drift rates for the target, flanker, and “other” (non-target/non-flanker) accumulators averaged over trials and models for congruent vs. incongruent trials. In case this was a source of confusion, we do not show the value of the flanker drift rates on congruent trials because the flanker and target accumulators are identical (i.e. the flanker/congruent drift rates are equivalent to the target/congruent drift rates).

(4) Lines 214-7: "The observation that single-unit information for target direction decreased between the fourth and final convolutional layers while population-level decoding remained high is especially noteworthy in that it implies a transition from representing target direction with specialized "target neurons" to a more distributed, ensemble-level code." Can the authors clarify why this is the only reasonable explanation for these results? It seems like many other explanations could be construed.

We have added additional clarification to this section and now use more tentative language:

“The observation that single-unit information for target direction decreased between the fourth and final convolutional layers indicates that the units become progressively less selective for particular target directions. Since population-level decoding remained high in these layers, this suggests a transition from representing target direction with specialized "target neurons" to a more distributed, ensemble-level code.”

(5) Lines 372-376: "Thus, simply training the model to perform the task is not sufficient to reproduce a behavioral phenomenon widely-observed in conflict tasks. This challenges a core (but often implicit) assumption of the task-optimized training paradigm, namely that to do a task well, a training model will result in model representations that are similar to those employed by humans." While I agree with the general sentiment, I feel that its application here is strange. Unless I'm missing something, in the context of the preceding sentence, the authors seem to be saying that researchers in the field expect that CNNs can produce a behavioral phenomenon (RTs) that is completely outside of their design and training. I don't think that anyone actually expects that.

We moved the discussion/analyses of RTs to the next paragraph. It should now be clear that this statement refers specifically to the absence of an accuracy congruency effect in the task-optimized models.

(6) Lines 387-389: "As a result, the VAMs may learn richer representations of the stimuli, since a variety of stimulus features-layout, stimulus position, flanker direction-influence behavior (Figure 2)." That is certainly true of tasks like this one where an optimal model would only focus on a tiny part of the image, whereas humans are distracted by many features. I'm not sure that this distractibility is the same as "richer representations". When CNNs classify images based on the background, would the authors claim that they have richer representations than humans?

We agree that “richer” may not be the best way to characterize these representations, and have changed it to “more complex”.

(7) Is it possible that drift rate d_k for each response happens to be negative on a given trial? If so, how is the decision given on such trials (since presumably none of the accumulators will ever reach the boundary)?

It is indeed possible for all of the drift rates to be negative, though we found that this occurred for a vanishingly small number of trials (mean ± s.e.m. percent trials/model: 0.080 ± 0.011%, n = 75 models), as reported in the Methods. These trials were excluded from analyses.

(8) Can the authors comment on how they chose the CNN architecture and whether they expect that different architectures will produce similar results?

Before establishing the seven-layer CNN architecture used throughout the paper, we conducted some preliminary experiments using other architectures that differed primarily in the number of CNN layers. We found that models with significantly fewer than seven layers typically failed to reach human-level accuracy on the task while larger models achieved human-level accuracy but (unsurprisingly) took longer to train.

**Reviewer #3 (Recommendations For The Authors)**:- In the introduction to this paper (particularly the paragraph beginning in line 33), the authors note that EAMs have typically been used in simplified settings and that they do not provide a means to account for how people extract information from naturalistic stimuli. While I agree with this, the idea of connecting CNNs of visual processing with EAMs for a joint modeling framework has been done. I recommend looking at and referencing these two articles as well as adjusting the tenor of this part of an introduction to better reflect the current state of the literature. For full disclosure, I am one of the authors on these articles. https://link.springer.com/article/10.1007/s42113-019-00042-1
https://www.sciencedirect.com/science/article/abs/pii/S0010027721001323

We agree—thanks for pointing this out. The revised Introduction now discusses prior related models in more detail (including those referenced above) and better clarifies the novel contributions of our model. We specifically highlight that a novel contribution of the VAM is that “the CNN and EAM parameters are jointly fitted to the RT, choice, and visual stimulus data from individual participants in a unified Bayesian framework.”

- The statement in lines 56-58 implies that this is the first article to glue CNNs together with EAMs. I would edit this accordingly based on the prior comment here and references provided. I will note that the second feature of the approach in this paper is still novel and really nice, namely the fact that the CNN and the EAM are jointly fitted. In the aforementioned references, the CNN is trained on the image set, and individual level Bayesian estimation was only applied to the EAM. Thus, it may be useful to highlight the joint estimation aspect of this investigation as well as how the uniqueness of the data available makes it possible.

Agreed—see above.

- Figure 3c and associated text. I understand the MI analysis you are performing here, however it is difficult to interpret as it stands. In the figure, what does a MI of 0.1 mean?? Can you give some context to that scale? I do find the interpretation of the hunchback shape in lines 210-222 to be somewhat of a stretch. The discussion that precedes (lines 199-209) this is clear and convincing. Can this discussion be strengthened more? And more interpretability of Figure 3c would be helpful; entropic scales can be hard to interpret without some context or scale associated.

The MI analyses in Fig. 3C (and also Figs. 4C and 6E) show normalized MI, in which the raw MI has been divided by the entropy of the stimulus feature distribution. This normalization facilitates comparing the MI for different stimulus features, which is relevant for Figs. 4C and 6E. The normalized MI has a possible range of [0, 1], where 1 indicates perfect correlation between the two variables and 0 indicates complete independence. We now note in the legend of these figures that the possible normalized MI range is [0, 1], which should help with interpreting these values. Our revised results section for Fig. 3C now also includes some additional remarks on our interpretation of the hunchback shape of the MI.

- Lines 244-248 and the analyses in Figure 3 suggest a change in the behavior of the CNN around layer 4. This is just a musing, but what would happen if you just used a 4 layer CNN, or even a 3 layer? This is not just a methods question. Your analysis suggests a transition from localized to distributed information representation. Right now, the EAM only sees the output of the distributed representation. What if it saw the results the more local representations from early layers? Of course, a shallower network may just form the distributed representations earlier, but it would interesting if there were a way to tease out not just the presence of distributed vs local representations, but the utility of those to the EAM.

Thanks for this interesting suggestion. We did do some preliminary experiments in models with fewer layers, though we only examined the outputs of these models and did not assess their representations. We found that models with 3–5 layers generally failed to achieve human-level accuracy on the task. In principle, one could relate this observation to the representations of these models as a means of assessing the relative utility of distributed/local representations. However, there are confounding factors that one would ideally control for in order to compare models with different numbers of layers in this fashion (namely, the number of parameters).

- Section Line 359 (Task optimized models) - It would be helpful to clarify here what these task-optimized models are being trained to do. As I understand it, they are being trained to directly predict the target direction. But are you asking them to learn to predict the true target direction? Or are you training them to predict what each individual responds? I think it is the second (since you have 75 of these), but it's not clear. I looked at the methods and still couldn't get a clear description of this. Also, are you just stripping the LBA off of the end of the CNN and then essentially putting a softmax in its place? If so, it would be helpful to say so.

The task-optimized models were actually trained to output the true target direction in each stimulus, rather than trained to match the decisions of the human participants. We trained 75 such models since we wanted to use exactly the same stimuli as were used to train each VAM. The task-optimized CNNs were identical to those used in the VAMs, except that the outputs of the last layer were converted to softmax-scored probabilities for each direction rather than drift rates. The Results and Methods section now included additional commentary that clarifies these points.

- Line 373-376: This statement is pretty well established at this point in the similarity judgement literature. I recommend looking at and referencing https://onlinelibrary.wiley.com/doi/full/10.1111/cogs.13226
https://www.nature.com/articles/s41562-020-00951-3
https://link.springer.com/article/10.1007/s42113-020-00073-z

Thanks for pointing this out. For reference, the statement in question is “Thus, simply training the model to perform the task is not sufficient to reproduce a behavioral phenomenon widely-observed in conflict tasks. This challenges a core (but often implicit) assumption of the task-optimized training paradigm, namely that training a model to do a task well will result in model representations that are similar to those employed by humans.”

We agree that the first and third reference you mention are relevant, and we now cite them along with some other relevant work. In our view, the second reference you mention is not particularly relevant (that paper introduces a new computational model for similarity judgements that is fit to human data, but does not comment on training models to perform tasks vs. fitting to human data).

- Line 387-388: "VAMs may learn richer representations". This is a bit of a philosophical point, but I'll go ahead and mention it. The standard VAM does not necessarily learn "richer" feature representations. Rather, you are asking the VAM and task-optimized models to do different things. As a result, they learn different representations. "Better" or "richer" is in the eye of the beholder. In one view, you could view the VAM performance as sub-par since it exhibits strange artifacts (congruency effects) and the expansion of dimensionality in the VAM representations is merely a side-effect of poor performance. I'm not advocating this view, just playing devils advocate and suggesting a more nuanced discussion of the difference between the VAM and task-optimized models.

We agree—this is a great point. We have changed this statement to read “the VAMs may learn more complex [rather than richer] representations of the stimuli”.

- Lines 567-570: Here you discuss how the LBA backend of the VAM can't account for shrinking spotlight-like RT effects but that fitting models to different RT quantiles helps overcome this. I find this to be one of the weakest points of the paper (the whole process of fitting RT quantiles separately to begin with). This is just a limitation of the RT component of the model. This is a great paper but this is just a limitation inherent in the model. I don't see a need to qualify this limitation and think it would be better to just point out that this is a limitation of the LBA itself (be more clear that it is the LBA that is the limiting factor here) and that this leaves room for future research. From your last sentence of this paragraph, I agree that recurrent CNNs would be interesting. I will note that RNN choice-RT models are out there (though not with CNNs as part of the model).

We agree and have revised this section of the Discussion accordingly (see our response to Reviewer #2 for more detail). We also removed the analyses of models trained on separate RT quantiles.